# EXecution-Eval: Can language models execute real-world code?

## Abstract

As language models advance, traditional benchmarks face challenges of dataset saturation and disconnection from real-world performance, limiting our understanding of true model capabilities. We introduce EXecution-Eval (EXE), a benchmark designed to assess LLMs' ability to execute code and predict program states. EXE attempts to address key limitations in existing evaluations: difficulty scaling, task diversity, training data contamination, and cost-effective scalability. Comprising over 30,000 tasks derived from 1,000 popular Python repositories on GitHub, EXE spans a wide range of lengths and algorithmic complexities. Tasks require models to execute code, necessitating various operations including mathematical reasoning, logical inference, bit manipulation, string operations, loop execution, and maintaining multiple internal variable states during computation. Our methodology involves: (a) selecting and preprocessing GitHub repositories, (b) generating diverse inputs for functions, (c) executing code to obtain ground truth outputs, and (d) formulating tasks that require models to reason about code execution. This approach allows for continuous new task generation for as few as 1,123 tokens, significantly reducing the risk of models "training on the test set." We evaluate several state-of-the-art LLMs on EXE, revealing insights into their code comprehension and execution capabilities. Our results show that even the best-performing models struggle with complex, multi-step execution tasks, highlighting specific computational concepts that pose the greatest challenges for today's LLMs. Furthermore, we review EXE's potential for finding and predicting errors to aid in assessing a model's cybersecurity capabilities. We propose EXE as a sustainable and challenging testbed for evaluating frontier models, offering insights into their internal mechanistic advancement.

## 1 Introduction

Language model benchmarks are facing challenges of rapid saturation (Ott et al., 2022) and an increasing disconnect from real-world performance perceived by end-users (Zheng et al., 2023). Due to this, benchmarks are being continually created to address failure modes; e.g. SuperGLUE targeting GLUE's low problem difficulty (Wang et al., 2019), BIG-bench targeting general low evaluation diversity (Srivastava et al., 2022) and Auto-Arena-Hard targeting training-set contamination and data diversity in Chatbot-Arena (Li et al., 2024)(Chiang et al., 2024). These failure modes all demonstrate the challenge in linking the mechanistic improvements within language models to human understandable tasks.

Hence, to maximise an evaluation's utility we aim to minimise the common failure modes of; a) difficulty, not ensuring an unbound scale of small trivial problems to complex multi-step problems, b) diversity, not ensuring a representative distribution across a large space of problems, c) novelty, not ensuring continually fresh, out-out-training data samples can be generated and, d) scalability, not ensuring tasks are cost-effective to generate in the thousands and beyond.

Motivated by these challenges we introduce EXecutionEval (EXE), an evaluation replicating one of the primary tasks humans perform while coding; predicting and comparing a final program state for a given set of inputs - seen in Figure 1. EXE is designed to avoid the aforementioned failure modes; emphasising difficulty (smooth scale from trivial 1-step, one-line functions to difficult 100s-of-step, multi-layer functions), diversity (unbound number of test cases generatable for tasks from

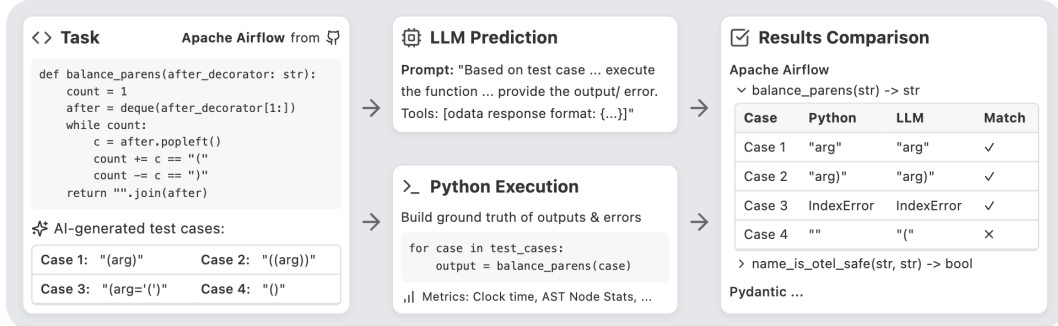

Figure 1: An example task from Apache Airflow's Github repository (code simplified to fit within diagram). EXE sources tasks from 1,000 Python repositories, generates test cases for them, and compares the LLM's ability to execute code against python's interpreter.

1,000 GitHub Repos), novelty (program inputs can be continually generated) and scalability (initial release containing 30,000+ problems at a cost of $33).

EXE also holds theoretical inspiration. (Fowler et al., 2022) et al have replicated positive pedagogical correlations found by (Lopez et al., 2008) between the abilities of CS1 students to "trace" programs (i.e. manually predict outputs and write the internal state out line by line) and their abilities to pass code writing and explanation exams. This is mirrored in CRUX-Eval's (Gu et al., 2024) findings, where they observe a moderate correlation between a model's ability to execute a block of code and a model's HumanEval (Chen et al., 2021) code writing Pass@1 rate.

## 2 EVALUATION FRAMEWORK

As seen in Figure 1, an EXE task is to predict a function's return value or error from: a) a code snippet and b) a set of input arguments. Code snippets are extracted from PyPi's most popular 1,000 python projects hosted on GitHub, we select our snippets to be pure (i.e. deterministic, no side effects), language model generatable (i.e. arg types of `ints`, `lists`, `...`) and to only require builtins (local imports and external libraries are inlined for the snippet). To realise this we follow the following three stage pipeline 2:

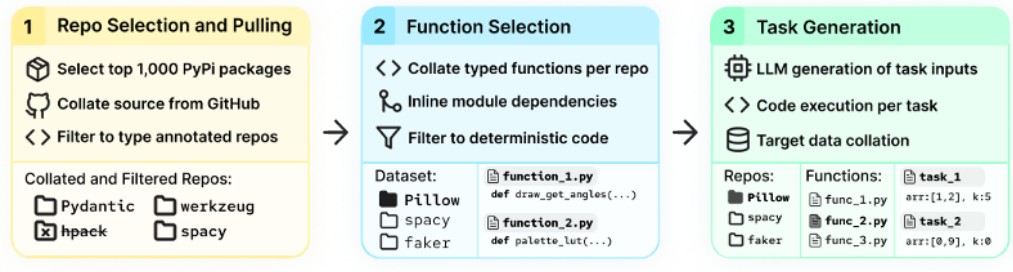

Figure 2: Three stage EXE task generation pipeline. Detailed example tasks and generated inputs can be found in Appendix A.1.

**1. Repo Selection and Code Scraping.** We first select the top 1,000 most popular pypi packages and collate the corresponding github repos where possible, similar to (Jimenez et al., 2023). Repositories are filtered to include only those with permissive licences that allow derivative works with attribution. These repos are then pulled down locally and filtered based on a static Abstract Syntax Tree (AST) analysis determining which repositories contain type-annotated code.

**2. Function Selection and Dependency Collation.** We perform a static AST analysis to filter to functions with LLM generatable argument and return type annotations. Further AST analysis

then recursively identifies dependent elements (modules, functions, classes, variables, ...) across files, builds a dependency graph, and inlines them into a base task. Finally, base tasks containing side effects or non-deterministic code such as environment variables, process calls, randomness or network requests are filtered out. See Appendix A.3 for step-by-step methodology and A.5 for detail on acceptable type annotations and filtering.

**3. Test Case Generation.** Using the argument type annotations we construct a LLM function calling schema that generates a diverse set of inputs. The base task code is then executed with each generated input and the result with runtime statistics are logged. This forms the test case (base task code + generated input), output (returned result or error from executed code) and statistics (runtime statistics + static AST analysis statistics). See Appendix A.2 for step-by-step methodology and Appendix A.6 for details on statistics.

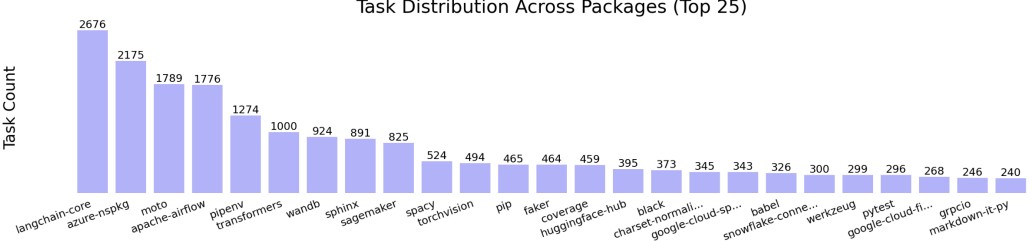

Figure 3: We observe task counts per repository to have a near logarithmic falloff. Note: Based on manual observations, several repositories are removed from EXE due to thousands of similar functions with only single modifications, for example changing a url address.

Through these stages of filtering, the original top 1,000 repositories are filtered down to the 33,875 task instances which comprise EXE. A high level breakdown of these task instances across repositories is presented in Figure 3. We note some repositories are overrepresented primarily due to being more modern (using type annotations) and the style of code (shorter deterministic pieces).

## 2.1 TASK FORMATION

**Model input.** The model is given a complete snippet of code alongside the input state to be executed. The model is then tasked to predict the resulting return value, or in the case that an exception is raised the model is instructed to generate an exception type and value. In practice, we prompt models with an odata json representation and use a parser to ensure valid generations. We do append one additional user reply with the parsing error if the model's response fails to parse. Examples of input instances can be found in Appendix A.1.

**Evaluation metrics.** To evaluate a proposed solution, we use the pass@k metric (Chen et al., 2021), comparing the ground truth and the generated prediction as json objects (`set` and `frozenset` are sorted before conversion to json lists). If the original code produced an exception, we compare the type and message (excluding stacktrace) using a language model comparison. See detailed methodology in Appendix A.7 and see examples of generated outputs in Appendix A.1.

## 2.2 FEATURES OF EXE

**Diversity of inputs and outputs.** Unlike many benchmarks focused on a particular subject matter area, a task in this eval may require a model to perform mathematical reasoning, logical inference, bit manipulation, string operations, loop execution, or to maintain multiple internal variables during computation. Furthermore, these may only form part of an algorithm that the model has to execute. Our random human inspection has uncovered algorithmic time complexities spanning from $O(1)$ to $O(x^n)$ and structured analysis has found tasks with code context lengths ranging from 440 to 311,000 tokens. Ensuring this broad diversity reduces the risk of hitting a local maxima and increases our opportunity to measure internal capabilities across a range of difficulties.

**Continually updatable.** Both our code collection and task input generation processes can create new tasks with minimal human oversight. Simply re-running our code collection to pull the latest

commits or directing it towards an uncollected Python GitHub repository will create new task instances. Furthermore we can continue to generate new test cases for existing tasks, our test case generator automatically avoids generating seen inputs. Hence, EXE can be extended continually with new task instances, ensuring answers were not included in training corpuses of models for evaluation.

**Cost effective scalability.** With generation of new tasks requiring an average of 1,112 input tokens (batch of 15) and evaluation of tasks typically requiring 1,123 tokens, ExecEval can be generated, tested and continually updated at a fraction of the cost of human-curated benchmarks. Our initial dataset of 33,875 cases has only incurred an approximate costing of $33 to produce and $95 to test on.

**Long multi-step problems with smooth difficulty scaling.** We provide a continuous spectrum of task difficulties, ranging from 1-step, one-line functions to multi-file, multi-class, multi-100-step tasks. Our most complex tasks include function call depths (non-recursive) of up to 13 levels (median: 2), separate identifier counts (i.e. variable names, function names, ...) of up to 823 (median: 16) and up to 63 if statements (median: 1). This smooth scaling of difficulty allows for a more detailed measurement of model coherence along multi-step problems than what is typically seen in traditional evaluations. However, as language models continue to advance rapidly, even this wide range of difficulties may eventually face saturation.

To address this, we observe a mechanism inspired by the SKILL-MIX evaluation (Yu et al., 2023) that leverages the typed nature of our function selection process. This approach allows us to create even more complex tasks by chaining functions where the output type of one matches the input type of another, or by combining multiple outputs into a composite input. The number of potential new tasks can be upper bounded by $n^2 \cdot (T_{\max})^k \cdot C$, where $n$ is the total number of types, $T_{\max} = \max_{i,j} T_{i,j}$ is the maximum number of existing tasks between any two types, $k$ is the number of functions to chain, and $C$ is the average number of test cases per task. While this is an upper bound and the actual number of valid composite tasks would be lower due to specific type compatibility constraints, it still represents a significant expansion of our task space. We view this as an opportunity to trade some of the 'realism' of using 100% real-world code for the ability to probe the upper bounds of model capabilities. For constant compute models, this approach allows us to test their internal mechanistic capabilities in handling increasingly complex, multi-step problems. And for chain-of-thought models, it provides a test of increasingly long-term agentic coherency.

**Error prediction.** To test the full spectrum of code execution we further generate test cases designed to trigger exceptions. Many of these require in-depth analysis to see ahead of time, for example predicting an invalid array index through multiple functions. While debugging exceptions is one of the more challenging software engineering tasks, we are yet to see it commonly evaluated in benchmarks.

## 3  RESULTS

We report our evaluation results across different SOTA models alongside our findings across different task statistics below.

Table 1: EXE Pass@1 results

| Model | EXE dataset (Pass@1) | Errors (Pass@1) |
|---|---|---|
| GPT-4o | 72.4 | 49.5 |
| GPT-4o-mini | 60.9 | 32.0 |
| Llama3.1-8B | 37.4 | 2.1 |
| Llama3.1-405B | 71.4 | 34.3 |
| Claude3.5-Sonnet | 76.1 | 45.8 |
| Mistral-Large-2407 | 71.5 | 33.7 |

**LLMs can execute real-world code, achieving results in-line with code generation benchmarks.** We find EXE shows similar relative model performance between models as seen in coding benchmarks such as HumanEval (Chen et al., 2021) and as seen in benchmarks requiring logical inference

such as (Lu et al., 2023). Furthermore we find a similar diversity of performance across packages as seen in agentic benchmarks such as (Jimenez et al., 2023). We show our findings in Figure 4.

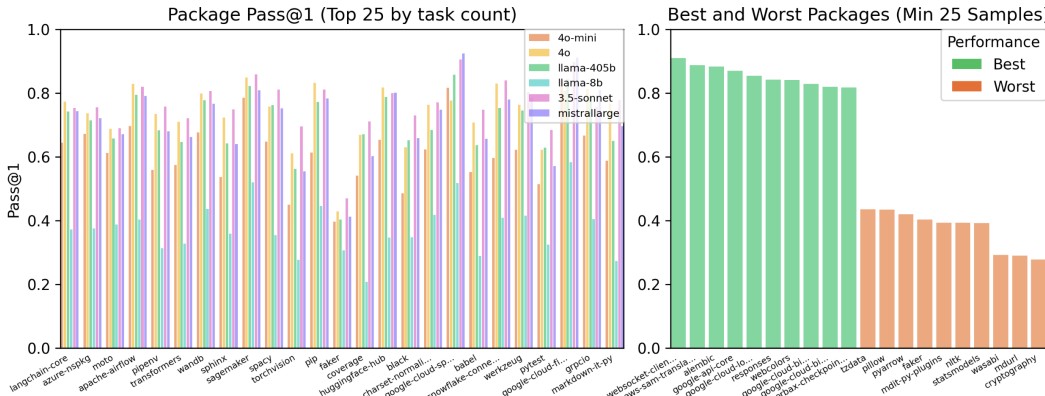

Figure 4: Left - We show the relative accuracy of different models across the top 20 packages by task count. Both the relative differences between models and the relative differences between packages are within expectations from other coding benchmarks (Jimenez et al., 2023). Right - We show the magnitude of diversity across packages (mean performance across all models).

Prior works such as Learning To Execute (Zaremba & Sutskever, 2014) and CRUX-Eval (Gu et al., 2024) have placed justifiable limitations on code complexity; removing mathematical operations, limiting line count, disallowing custom classes and only having one singular function to name a few. We hypothesised that these are no longer necessary, and to understand the true internal capabilities of a constant compute model (i.e. no Chain of Thought) we must test on real-world code, only applying limitations where forced (i.e. no arbitrary object inputs, as LLMs can't generate them). Our results as seen in table 1 provide initial evidence towards our hypothesis.

**ExecEval provides a smooth curve of task difficulties.** We set out to ensure a) our evaluation does not induce saturation from a bounded distribution of task difficulties, b) our evaluation does not induce an "AI overhang" by not having a smooth transition between difficulties and, c) the correlated factors affecting difficulty are human interpretable.

As shown in Figure 5 several task statistics such as "lines of code", "processing time" and "number of function calls" all correlate log-linearly with a model's achieved pass@1 score. These correlations provide preliminary evidence towards c) as they align with simplistic human intuition, i.e. more lines of code, more compute cycles, higher difficulty. Furthermore, we view the log-linear relationships as evidence towards b), i.e. EXE provides a smooth transition between difficulties. And finally, we view the relationships as a demonstration of difficulty being affected by factors within our control, i.e. number of function calls - providing empirical evidence towards a).

Beyond evaluation-wide difficulty scaling, EXE also demonstrates diversity and varying difficulty levels within individual task sets. Each function has up to 15 generated test cases, allowing us to analyse variance per task set. To measure execution path diversity, we collect runtime statistics (detailed in Appendix A.6) and find a mean Coefficient of Variation (CV) of 0.61 for "Count of conditionals executed", indicating substantial variation in code paths taken. Furthermore we find a CV of 0.20 for "lines executed", showing significant diversity in the number of steps required to answer. Finally, we measure diversity in generated task difficulty through model performance - GPT-4o achieves a mean pass rate of 0.742 ($\sigma = 0.293$) per function, providing empirical evidence test cases present a difficulty scale.

**ExecEval's test case generation scales.** While EXE today includes up to 15 test cases per task, our analysis demonstrates EXE's generation pipeline can scale significantly further without plateauing. As shown in Figure 6, generation of novel test case continues well beyond 300 cases per task while maintaining all quality controls (detailed in Appendix A.2) - implying a potential dataset scale-up lower bound of 20x. Growth rates vary across specific functions - for example, langchain-core's image formatting function, which requests a base64 encoded image string, shows the lowest growth

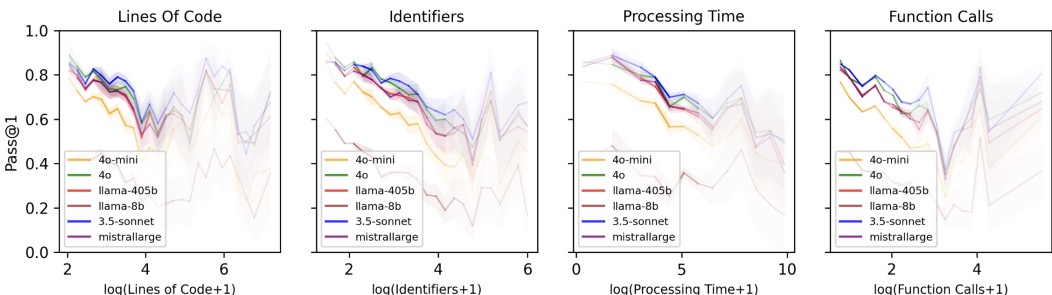

Figure 5: Pass@1 for all tasks across four of our code metrics. The shaded area represents variance, and the opacity is scaled with count of samples. Processing time is measured in microseconds.

rate. This aligns with intuition - generating novel, base64 images poses significantly more difficulty than generating diverse string or numeric inputs.

Importantly, our token efficiency analysis (right plot) reveals that significant scaling is possible without proportional prompt growth. By randomly selecting and injecting just 60 prior cases into the generation prompt, we can effectively generate over 1,000 novel cases. This sublinear token growth suggests the potential for substantial dataset expansion without incurring prohibitive costs. Detailed examples of tasks and their generated test cases are provided in Appendix A.8.

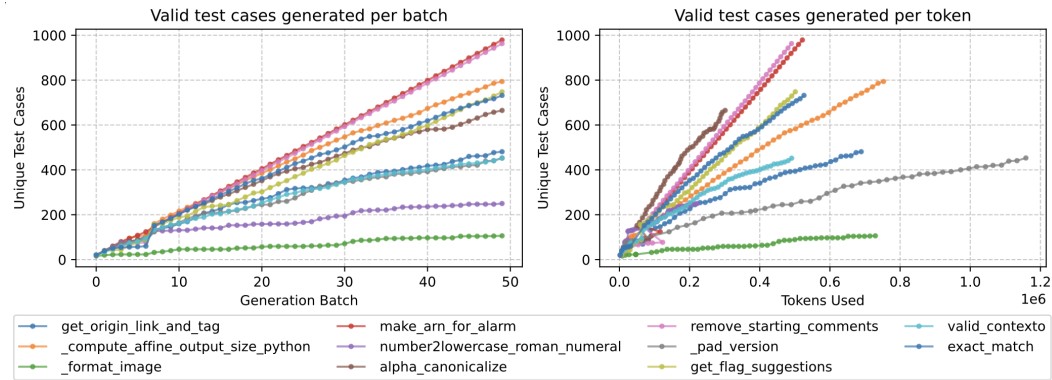

Figure 6: Test case generation analysis across eleven diverse Python functions sourced from popular libraries including Azure, PyTorch, Langchain, and NLTK. Functions range from geometric computations (torchvision) to SQL regex (snowflake-python-connector). Left: Cumulative unique validated test cases per generation batch. Right: Same data plotted against token usage, showing generation cost is largely constant per batch (primary factor is initial task code length). Further methodology and source code for tested functions are provided in Appendix A.8.

**Stylistic coding patterns shape the metrics.** As can be seen in Figure 5 the pass@1 rate of function calls hits an elbow and then surprisingly improves as the call count increases. During our investigation we found several of these occurrences, and not only with call count. These were found to be largely driven by specific coding patterns and complex tasks that LLMs excel at. We show in Figure 7 below three example tasks, and more specifically coding patterns driving this anomaly.

**LLMs struggle with certain coding features.** As EXE contains a diverse set of tasks, we are able to observe model performance differing greatly based on coding features used in any task. To illustrate: floating point math operations such as multiplications (GPT-4o: 43 mean Pass@1) significantly increase task difficulty, however bit manipulation and boolean operations only showed a minor negative impact. Iterative operations such as compound assignment operations i.e. "i += 1" (56 Pass@1), list slicing (65 Pass@1) and list comprehensions (68 Pass@1) all increased difficulty, however for loops on (73 Pass@1) on average did not have a significant impact.

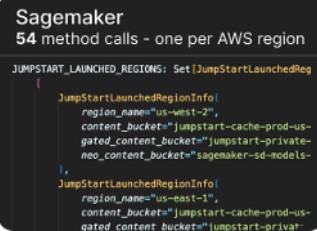

Figure 7: Three examples of high pass@1 rate tasks that contain large amounts of function calls. Left - Charset-normaliser performs 300+ function calls to define ranges of unicode characters upon initialisation; this constant has little effect on task difficulty but is used frequently and hence appears in many tasks. Middle - Langchain's Unparser class traverses an AST and regenerates source code. The calling method in our dataset is "add_last_line_print(str) → str" which takes in code, parses it and then uses Unparse(...) to unparse it; this is a prime example of a "directly predictable task", i.e. one not requiring line by line code execution to predict a result. Right - Similar to Charset-normaliser, AWS's Sagemaker has a module level constant with 10s of calls; not creating a large impact on task difficulty but frequent in its use.

With the above metrics, and those seen in Figure 7, their mean Pass@k decreases as their count increases. To reduce the risk of our metrics being a proxy for longer problems we show the effects can still be seen below in Figure 8 after normalisation by lines of code (only lines with executable syntax tokens are counted).

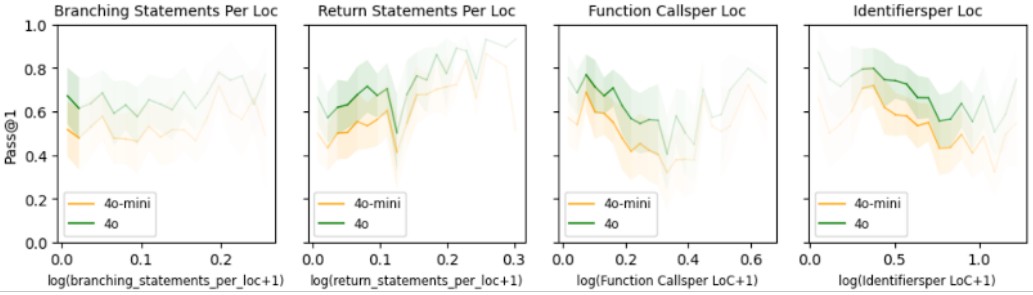

Figure 8: Pass@1for all tasks across four of our code metrics normalised by line of code count (limited to GPT models for readability). All four of the above metrics previously showed a negative impact as they increased, interestingly we now observe branching statements having little to no impact and return statements surprisingly driving an increase in Pass@1 score. Our strong negative factors i.e. function calls and identifiers created, still are seen increasing task difficulty as they take up ever greater percentages of the task.

## 4 RELATED WORK

There is a rich history of work on evaluating language models' abilities in reasoning, execution, and multi-step problem-solving across various domains. These efforts span from natural language processing to mathematical reasoning, and from code generation to program execution. Our work, EXecution-Eval (EXE), builds upon this foundation while addressing key challenges in benchmark design and evaluation.

Code generation benchmarks have been the foundation of evaluating the coding abilities of language models. Works like HumanEval (Chen et al., 2021) and MBPP (Austin et al., 2021) established standardised datasets for assessing code synthesis from natural language descriptions. These efforts have expanded to cover multiple programming languages (Cassano et al., 2022; Khan et al., 2023) and more complex domains such as algorithmic problem solving (Huang et al., 2023). While these benchmarks focus primarily on the task of code generation, we believe additional focus on the tasks of code execution and error prediction have been overlooked and may offer additional insight into the internal capabilities of frontier models.

The concept of "learning to execute" itself has a long history, Zaremba & Sutskever (2014) explored neural networks' ability to learn and execute simple programs. Graves et al. (2014) constructed the first Neural Turing Machines with (Kaiser & Sutskever, 2015; Reed & de Freitas, 2015; Dehghani et al., 2018) all building further into this domain. This line of research has evolved, with recent works like Bieber et al. (2020); Nye et al. (2021) and Gu et al. (2024) applying graph and language models to execute synthetic or simplistic Python programs. EXE builds upon these foundations by evaluating execution capabilities on complex, messy, real-world code from diverse GitHub repositories, providing a more challenging, scaleable and realistic test bed.

Recent trends in benchmark design have emphasised the importance of diverse, multi-step problems and agentic capabilities. Works like Jimenez et al. (2023) have introduced benchmarks that require solving real world software engineering problems while Zhou et al. (2023) has enabled evaluation of complex instruction following and performing multi-step reasoning. In the mathematical domain, benchmarks like those by Hendrycks et al. (2021) and Lu et al. (2023) have pushed models to solve intricate, multi-step problems.

The challenge of benchmark saturation and the need for continually updated evaluations has been recognized in recent works (Ott et al., 2022). Live benchmarks such as those proposed by Li et al. (2024), (Chiang et al., 2024) and Kiela et al. (2021) aim to address this issue. Skill-Mix (Yu et al., 2023) takes a novel approach, combining separate skills required to solve a problem they are able to increase task difficulty non-linearly with $k$ skills. EXE has been inspired by both these concepts, hence the focus on enabling continual generation of new coding tasks and test cases, as well as the potential extension into chaining functions.

While many existing benchmarks use curated or synthetic datasets, EXE leverages real-world code from popular Python repositories. This approach is inspired by works like CodeNet (Puri et al., 2021) and The Stack (Kocetkov et al., 2022) which demonstrated the value of diverse, real-world data in training and evaluating language models.

## 5 EXTENSIONS

**Expanding the scope and diversity** We believe scaling EXE to include more repositories by as much as 100x would significantly reduce the noise seen in our coding metrics and provide a more resilient baseline for future frontier models. By incorporating additional Python functions — potentially using language models to predict missing type annotations — and including a diversity of other programming languages such as C++, Go and JavaScript, we believe there is even further opportunity to scale. This would offer further insights into the generalisability of a model's code understanding, pose new challenges for analysis such as pointers, macros and type-free codebases.

**Probing code execution mechanisms with simple functions** We believe there is an opportunity to align code execution with mechanistic interpretability, to gain an understanding of how constant compute language models can execute complex multi-step instructions. To illustrate, if we select the simplest function that a language model can not directly predict the outcome of, a hash function for example (one that doesn't use floating point math in this case), one requiring compute at each iteration. This would force the network to perform the computation step by step, and for a constant compute feed-forward network, layer by layer. Hence, performing a single iteration that may not lead to anything interesting, however as we increase the iteration count one by one, the model now must find a repeated circuit to perform the same computation in the later layers. For every increase it must find another circuit or a more optimal way of performing its work until it fails. We believe this would present an interesting approach alongside standard mechanistic interpretability techniques for circuit discovery and understanding of control flow, variable tracking and computational logic at the mechanistic level.

**Breakpoint analysis for validating code execution granularly** Rather than evaluating the final return value, including multiple evaluation points within code execution may assist verification of if models are performing the step-by-step computations to reach a return value. Furthermore by inserting 'breakpoints' throughout the execution process, we can transform a single return state prediction task into numerous intermediate state prediction tasks. To illustrate, given a code snippet with a breakpoint at a specific line, a model would be tasked to determine the values of the local variables when the breakpoint is triggered. This mirrors common human debugging practices and may reveal

discrepancies between final output accuracy and intermediate state understanding, offering further resistance against tasks where their final outcome can be directly predicted.

**Connection to cybersecurity threat model.** Software vulnerability research techniques are largely [1] enabled by the ability to predict and reason about expected program outcomes. For example, code injection, path resolution and memory buffer attacks are often found through manual human analysis; tracing inputs through the control flow, predicting output states and reasoning if there are opportunities to exploit. As EXE contains parsers such as seen in Appendix A.1 we see an opportunity to select a subset of EXE where prediction of error would imply language models have the internal capability to comprehend and aid humans with crafting vulnerabilities.

## 6 CONCLUSIONS

In this paper, we introduced EXecution-Eval (EXE), a benchmark designed to evaluate whether language models can execute real-world code. By collecting over 30,000 tasks from 1,000 popular Python repositories, EXE presents a diverse range of problems requiring computational operations such as mathematical reasoning, logical inference, and state maintenance. Our evaluations suggest that while language models demonstrate some capability in executing code, they often struggle with complex, multi-step tasks—particularly those involving many identifiers, function calls and iterative operations. Our findings indicate that although current models have limitations in accurately reasoning about and executing real-world code, they perform surprisingly well on average, prompting several opportunities extending this investigation.

EXE aims to address limitations of existing benchmarks by providing a scalable, diverse, and continually updatable framework. Its design targets a smooth difficulty scale and easy generation of new tasks with minimal human oversight with the goal to reduce the risk of models "training on the test set."

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

## A  APPENDIX

You may include other additional sections here.

### A.1  EXAMPLE INPUT & OUTPUT

Below is an example from the evaluation set. It is split into three components:

**1. Code Task.** The function `split_email` was found to pass the type requirements, and as such all modules, classes, functions and attributes required to execute it have been recursively inlined.

**2. Test Case Inputs.** Based on the type definition (used for setting the function calling schema) inputs/ output pairs have been generated with the goal of maximising diversity of control flow paths within the function.

**3. Outputs.** Based on the type definition (used for setting the function calling schema) inputs/ output pairs have been generated with the goal of maximising diversity of control flow paths within the function.

**Examples**

### A.1.1 EXAMPLE A.

**Code**

Note: The top 1,000 PyPI repos are used to form EXE, this function is from celery, rank 594

```python
def abbr(S: str, max: int, ellipsis: str | bool = '...') -> str:
    """Abbreviate word."""
    if S is None:
        return '???'
    if len(S) > max:
        return isinstance(ellipsis, str) and (
            S[: max - len(ellipsis)] + ellipsis) or S[: max]
    return S

def abbrtask(S: str, max: int) -> str:
    """Abbreviate task name."""
    if S is None:
        return '???'
    if len(S) > max:
        module, _, cls = S.rpartition('.')
        module = abbr(module, max - len(cls) - 3, False)
        return module + '[.]' + cls
    return S
```

**Test Case Inputs**

Note: For quick groking, only three inputs are shown for this example. Standard tasks contain 15 generated inputs.

```
[
    {
        "input": [["module.ClassName",15], {}],
        "output": "mod[.]ClassName",
    },
    {
        "input": [["long.module.name.with.many.parts.ClassName",25], {}],
        "output": "long.module.n[.]ClassName",
    },
    {
        "input": [["module.ClassName", 3], {}],
        "output": "[.]ClassName",
    },
]
```

**Generated Outputs**

```
[
    {
        "input": [["module.ClassName",15], {}],
        "output": "mod[.]ClassName",
        "prediction": "module[.]ClassName",
        "result": false,
        "answer_tokens": {"completion": 15, "prompt": 781, "total": 796}
    },
    {
        "input": [["long.module.name.with.many.parts.ClassName",25], {}],
        "output": "long.module.n[.]ClassName",
        "prediction": "long.module.name[.]ClassName",
        "result": false,
        "answer_tokens": {"completion": 17, "prompt": 787, "total": 804}
    },
```

```
    {
        "input": [["module.ClassName", 3], {}],
        "output": "[.]ClassName",
        "prediction": "[.]ClassName",
        "result": true,
        "answer_tokens": {"completion": 14, "prompt": 781, "total": 795}
    },
]
```

### A.1.2 EXAMPLE B.

**Code**

This function is from email-validator, rank 345.

```python
from typing import Optional, Tuple
import re
import unicodedata

class EmailNotValidError(ValueError):
    """Parent class of all exceptions raised by this module."""
    pass

class EmailSyntaxError(EmailNotValidError):
    """Exception raised when an email address fails validation because
    ↪  of its form."""
    pass

ATEXT = r'a-zA-Z0-9_!#\$%&\'\*\+\-/=\?\^`\{\|\}~'

def safe_character_display(c: str) -> str:
    # Return safely displayable characters in quotes.
    if c == '\\':
        return f"\"{c}\""   # can't use repr because it escapes it
    if unicodedata.category(c)[0] in ("L", "N", "P", "S"):
        return repr(c)

    # Construct a hex string in case the unicode name doesn't exist.
    if ord(c) < 0xFFFF:
        h = f"U+{ord(c):04x}".upper()
    else:
        h = f"U+{ord(c):08x}".upper()

    # Return the character name or, if it has no name, the hex string.
    return unicodedata.name(c, h)

ATEXT_RE = re.compile('[.' + ATEXT + ']')  # ATEXT plus dots

def check_unsafe_chars(s: str, allow_space: bool = False) -> None:
    # Check for unsafe characters or characters that would make the
    ↪  string
    # invalid or non-sensible Unicode.
    bad_chars = set()
    for i, c in enumerate(s):
```

```python
            category = unicodedata.category(c)
            if category[0] in ("L", "N", "P", "S"):
                # Letters, numbers, punctuation, and symbols are permitted.
                pass
            elif category[0] == "M":
                # Combining character in first position would combine with
                ↪   something
                # outside of the email address if concatenated, so they are
                ↪   not safe.
                # We also check if this occurs after the @-sign, which would
                ↪   not be
                # sensible because it would modify the @-sign.
                if i == 0:
                    bad_chars.add(c)
            elif category == "Zs":
                # Spaces outside of the ASCII range are not specifically
                ↪   disallowed in
                # internationalized addresses as far as I can tell, but they
                ↪   violate
                # the spirit of the non-internationalized specification that
                ↪   email
                # addresses do not contain ASCII spaces when not quoted.
                ↪   Excluding
                # ASCII spaces when not quoted is handled directly by the
                ↪   atom regex.
                #
                # In quoted-string local parts, spaces are explicitly
                ↪   permitted, and
                # the ASCII space has category Zs, so we must allow it here,
                ↪   and we'll
                # allow all Unicode spaces to be consistent.
                if not allow_space:
                    bad_chars.add(c)
            elif category[0] == "Z":
                # The two line and paragraph separator characters (in
                ↪   categories Zl and Zp)
                # are not specifically disallowed in internationalized
                ↪   addresses
                # as far as I can tell, but they violate the spirit of the
                ↪   non-internationalized
                # specification that email addresses do not contain line
                ↪   breaks when not quoted.
                bad_chars.add(c)
            elif category[0] == "C":
                # Control, format, surrogate, private use, and unassigned
                ↪   code points (C)
                # are all unsafe in various ways. Control and format
                ↪   characters can affect
                # text rendering if the email address is concatenated with
                ↪   other text.
                # Bidirectional format characters are unsafe, even if used
                ↪   properly, because
                # they cause an email address to render as a different email
                ↪   address.
                # Private use characters do not make sense for publicly
                ↪   deliverable
                # email addresses.
                bad_chars.add(c)
            else:
                # All categories should be handled above, but in case there
                ↪   is something new
                # to the Unicode specification in the future, reject all
                ↪   other categories.
                bad_chars.add(c)
```

```
756
757             if bad_chars:
758                 raise EmailSyntaxError("The email address contains unsafe
                    ↪    characters: "
759                                     + ", ".join(safe_character_display(c) for
760                                     ↪   c in sorted(bad_chars)) + ".")
761
762
763     def split_email(email: str) -> Tuple[Optional[str], str, str, bool]:
764         # Return the display name, unescaped local part, and domain part
765         # of the address, and whether the local part was quoted. If no
766         # display name was present and angle brackets do not surround
767         # the address, display name will be None; otherwise, it will be
768         # set to the display name or the empty string if there were
769         # angle brackets but no display name.
770
771         # Typical email addresses have a single @-sign and no quote
772         # characters, but the awkward "quoted string" local part form
773         # (RFC 5321 4.1.2) allows @-signs and escaped quotes to appear
774         # in the local part if the local part is quoted.
775
776         # A `display name <addr>` format is also present in MIME messages
777         # (RFC 5322 3.4) and this format is also often recognized in
778         # mail UIs. It's not allowed in SMTP commands or in typical web
779         # login forms, but parsing it has been requested, so it's done
780         # here as a convenience. It's implemented in the spirit but not
781         # the letter of RFC 5322 3.4 because MIME messages allow newlines
782         # and comments as a part of the CFWS rule, but this is typically
                ↪    not
783         # allowed in mail UIs (although comment syntax was requested once
                ↪    too).
784         #
785         # Display names are either basic characters (the same basic
                ↪    characters
786         # permitted in email addresses, but periods are not allowed and
                ↪    spaces
787         # are allowed; see RFC 5322 Appendix A.1.2), or or a quoted string
                ↪    with
788         # the same rules as a quoted local part. (Multiple quoted strings
                ↪    might
789         # be allowed? Unclear.) Optional space (RFC 5322 3.4 CFWS) and
                ↪    then the
790         # email address follows in angle brackets.
791         #
792         # We assume the input string is already stripped of leading and
                ↪    trailing CFWS.
793
794         def split_string_at_unquoted_special(text: str, specials:
            ↪    Tuple[str, ...]) -> Tuple[str, str]:
795             # Split the string at the first character in specials (an
                ↪    @-sign
796             # or left angle bracket) that does not occur within quotes and
797             # is not followed by a Unicode combining character.
798             # If no special character is found, raise an error.
799             inside_quote, escaped, left_part = False, False, ""
800             for i, c in enumerate(text):
801                 # < plus U+0338 (Combining Long Solidus Overlay) normalizes
                    ↪    to
802                 #  U+226E (Not Less-Than), and  it would be confusing to
                    ↪    treat
803                 # the < as the start of "<email>" syntax in that case.
                    ↪    Likewise,
```

```python
            # if anything combines with an @ or ", we should probably
            ↪  not
            # treat it as a special character.
            if unicodedata.normalize("NFC", text[i:])[0] != c:
                left_part += c

        elif inside_quote:
                left_part += c
                if c == '\\' and not escaped:
                    escaped = True
                elif c == '"' and not escaped:
                    # The only way to exit the quote is an unescaped
                    ↪  quote.
                    inside_quote = False
                    escaped = False
                else:
                    escaped = False
        elif c == '"':
                left_part += c
                inside_quote = True
        elif c in specials:
                # When unquoted, stop before a special character.
                break
        else:
                left_part += c

    if len(left_part) == len(text):
        raise EmailSyntaxError("An email address must have an
        ↪  @-sign.")

    right_part = text[len(left_part):] # The right part is whatever
    ↪  is left.

    return left_part, right_part

def unquote_quoted_string(text: str) -> Tuple[str, bool]:
    # Remove surrounding quotes and unescape escaped backslashes
    # and quotes. Escapes are parsed liberally. I think only
    ↪  backslashes
    # and quotes can be escaped but we'll allow anything to be.
    quoted, escaped, value = False, False, ""
    for i, c in enumerate(text):
        if quoted:
            if escaped:
                value += c
                escaped = False
            elif c == '\\':
                escaped = True
            elif c == '"':
                if i != len(text) - 1:
                    raise EmailSyntaxError("Extra character(s) found
                    ↪  after close quote: "
                                        + ",
                                        ↪  ".join(safe_character_display(c)
                                        ↪  for c in text[i + 1:]))
                break
            else:
                value += c
        elif i == 0 and c == '"':
```

```
864
865                    quoted = True
866               else:
867                    value += c
868
869          return value, quoted
870
871
872      # Split the string at the first unquoted @-sign or left angle
      ↪   bracket.
873      left_part, right_part = split_string_at_unquoted_special(email,
874      ↪   ("@", "<"))
875
876      # If the right part starts with an angle bracket, then the left
877      ↪   part
878      # is a display name and the rest of the right part up to the
879      # final right angle bracket is the email address, .
880      if right_part.startswith("<"):
881          # Remove space between the display name and angle bracket.
882          left_part = left_part.rstrip()
883
884          # Unquote and unescape the display name.
885          display_name, display_name_quoted =
886          ↪   unquote_quoted_string(left_part)
887
888          # Check that only basic characters are present in a non-quoted
889          ↪   display name.
890          if not display_name_quoted:
891              bad_chars = {
892                      safe_character_display(c)
893                      for c in display_name
894                      if (not ATEXT_RE.match(c) and c != ' ') or c == '.'
895              }
896              if bad_chars:
897                      raise EmailSyntaxError("The display name contains
898                      ↪   invalid characters when not quoted: " + ",
899                      ↪   ".join(sorted(bad_chars)) + ".")
900
901          check_unsafe_chars(display_name, allow_space=True) # Check for
902          ↪   other unsafe characters.
903
904          # Check that the right part ends with an angle bracket
905          # but allow spaces after it, I guess.
906          if ">" not in right_part:
907              raise EmailSyntaxError("An open angle bracket at the start
908              ↪   of the email address has to be followed by a close angle
909              ↪   bracket at the end.")
910          right_part = right_part.rstrip(" ")
911          if right_part[-1] != ">":
912              raise EmailSyntaxError("There can't be anything after the
913              ↪   email address.")
914
915          # Remove the initial and trailing angle brackets.
916          addr_spec = right_part[1:].rstrip(">")
917
          # Split the email address at the first unquoted @-sign.
          local_part, domain_part =
          ↪   split_string_at_unquoted_special(addr_spec, ("@",))
```

```
        # Otherwise there is no display name. The left part is the local
        # part and the right part is the domain.
        else:
            display_name = None
            local_part, domain_part = left_part, right_part

        if domain_part.startswith("@"):
            domain_part = domain_part[1:]

        # Unquote the local part if it is quoted.
        local_part, is_quoted_local_part =
        ↪  unquote_quoted_string(local_part)

        return display_name, local_part, domain_part, is_quoted_local_part
```

**Test Case Inputs**

```
[
    {
        "input": [["simple@example.com"], {}],
        "output": [null,"simple","example.com", false],
    },
    {
        "input": [["user+name@sub.domain.com"], {}],
        "output": [null,"user+name","sub.domain.com", false],
    },
    {
        "input": [["user.name@domain.co.uk"], {}],
        "output": [null,"user.name","domain.co.uk", false],
    },
    {
        "input": [["\"quoted@local\"@example.com"], {}],
        "output": [null,"quoted@local","example.com", true],
    },
    {
        "input": [["display name <user@domain.com>"], {}],
        "output": ["display name","user","domain.com", false],
    },
    {
        "input": [["user@localhost"], {}],
        "output": [null,"user","localhost", false],
    },
    {
        "input": [["user@[IPv6:2001:db8::1]"], {}],
        "output": [null,"user","[IPv6:2001:db8::1]", false],
    },
    {
        "input": [["\"escaped\\\"quote\"@example.com"], {}],
        "output": [null,"escaped\"quote","example.com", true],
    },
    {
        "input": [["user.name@longsubdomain.example.com"], {}],
        "output": [null,"user.name","longsubdomain.example.com", false],
    },
    {
        "input": [["very.common@example.com"], {}],
        "output": [null,"very.common","example.com", false],
    },
```

```
    {
        "input": [["user@domain-with-dash.com"], {}],
        "output": [null,"user","domain-with-dash.com", false],
    },
    {
        "input": [["user@123.123.123.123"], {}],
        "output": [null,"user","123.123.123.123", false],
    },
    {
        "input": [["\"much.more unusual\"@example.com"], {}],
        "output": [null,"much.more unusual","example.com", true],
    },
    {
        "input": [["user@xn--exmple-cua.com"], {}],
        "output": [null,"user","xn--exmple-cua.com", false],
    },
    {
        "input": [["user@domain_with_underscore.com"], {}],
        "output": [null,"user","domain_with_underscore.com", false],
    }
]
```

**Generated Outputs**

```
[
    {
        "input": [["simple@example.com"], {}],
        "output": [null,"simple","example.com", false],
        "prediction": [null,"simple","example.com",false],
        "result": true,
        "answer_tokens": {"completion": 18,"prompt": 4610,"total": 4628}
    },
    {
        "input": [["user+name@sub.domain.com"], {}],
        "output": [null,"user+name","sub.domain.com", false],
        "prediction": [null,"user+name","sub.domain.com",false],
        "result": true,
        "answer_tokens": {"completion": 21,"prompt": 4614,"total": 4635}
    },
    {
        "input": [["user.name@domain.co.uk"], {}],
        "output": [null,"user.name","domain.co.uk", false],
        "prediction": [null,"user.name","domain.co.uk",false],
        "result": true,
        "answer_tokens": {"completion": 20,"prompt": 4613,"total": 4633}
    },
    {
        "input": [["\"quoted@local\"@example.com"], {}],
        "output": [null,"quoted@local","example.com", true],
        "prediction": ["null","quoted@local","example.com",true],
        "result": false,
        "answer_tokens": {"completion": 20,"prompt": 4615,"total": 4635}
    },
    {
        "input": [["display name <user@domain.com>"], {}],
        "output": ["display name","user","domain.com", false],
        "prediction": ["display name","user","domain.com",false],
        "result": true,
        "answer_tokens": {"completion": 19,"prompt": 4615,"total": 4634}
    },
    {
        "input": [["user@localhost"], {}],
        "output": [null,"user","localhost", false],
```

```
        "prediction": [null,"user","localhost",false],
        "result": true,
        "answer_tokens": {"completion": 17,"prompt": 4610,"total": 4627}
    },
    {
        "input": [["user@[IPv6:2001:db8::1]"], {}],
        "output": [null,"user","[IPv6:2001:db8::1]", false],
        "prediction": "EmailSyntaxError: An email address must have an
        ↪  @-sign.",
        "result": false,
        "answer_tokens": {"completion": 24,"prompt": 4620,"total": 4644
        }
    },
    {
        "input": [["\"escaped\\\"quote\"@example.com"], {}],
        "output": [null,"escaped\"quote","example.com", true],
        "prediction": ["null","escaped\"quote","example.com",true],
        "result": false,
        "answer_tokens": {"completion": 20,"prompt": 4615,"total": 4635}
    },
    {
        "input": [["user.name@longsubdomain.example.com"], {}],
        "output": [null,"user.name","longsubdomain.example.com", false],
        "prediction":
        ↪  [null,"user.name","longsubdomain.example.com",false],
        "result": true,
        "answer_tokens": {"completion": 22,"prompt": 4615,"total": 4637}
    },
    {
        "input": [["very.common@example.com"], {}],
        "output": [null,"very.common","example.com", false],
        "prediction": [null,"very.common","example.com",false],
        "result": true,
        "answer_tokens": {"completion": 19,"prompt": 4611,"total": 4630}
    },
    {
        "input": [["user@domain-with-dash.com"], {}],
        "output": [null,"user","domain-with-dash.com", false],
        "prediction": [null,"user","domain-with-dash.com",false],
        "result": true,
        "answer_tokens": {"completion": 21,"prompt": 4614,"total": 4635}
    },
    {
        "input": [["user@123.123.123.123"], {}],
        "output": [null,"user","123.123.123.123", false],
        "prediction": [null,"user","123.123.123.123",false],
        "result": true,
        "answer_tokens": {"completion": 23,"prompt": 4616,"total": 4639}
    },
    {
        "input": [["\"much.more unusual\"@example.com"], {}],
        "output": [null,"much.more unusual","example.com", true],
        "prediction": [null,"much.more unusual","example.com",true],
        "result": true,
        "answer_tokens": {"completion": 20,"prompt": 4615,"total": 4635}
    },
    {
        "input": [["user@xn--exmple-cua.com"], {}],
        "output": [null,"user","xn--exmple-cua.com", false],
        "prediction": [null,"user","xn--exmple-cua.com",false],
        "result": true,
        "answer_tokens": {"completion": 24,"prompt": 4617,"total": 4641}
    },
    {
```

```
     "input": [["user@domain_with_underscore.com"], {}],
     "output": [null,"user","domain_with_underscore.com", false],
     "prediction": "EmailSyntaxError: The email address contains unsafe
     ↪  characters: 'U+005F'.",
     "result": false,
     "answer_tokens": {"completion": 28,"prompt": 4614,"total": 4642}
   }
]
```

## A.2 INPUT GENERATION

Test case generation is performed through a three-stage pipeline: schema construction, test generation, and validation.

### A.2.1 SCHEMA CONSTRUCTION

Using our AST analysis's findings (see Section A.5), we construct OpenAPI-compatible JSON schemas from identified argument and return types. Consider a type-annotated function from our dataset:

```python
from typing import Dict, List, Optional, Tuple, Union

def get_tree_starting_at(module: str, edges: List[Tuple[str, str]]) ->
↪  List[Union[str, List[str]]]:
    """
    Returns the tree starting at a given module following all edges.

    Args:
        module (`str`): The module that will be the root of the subtree
        ↪  we want.
        eges (`List[Tuple[str, str]]`): The list of all edges of the
        ↪  tree.

    Returns:
        `List[Union[str, List[str]]]`: The tree to print in the
        ↪  following format: [module, [list of edges
        starting at module], [list of edges starting at the preceding
        ↪  level], ...]
    """
    vertices_seen = [module]
    new_edges = [edge for edge in edges if edge[0] == module and edge[1]
    ↪  != module and "__init__.py" not in edge[1]]
    tree = [module]
    while len(new_edges) > 0:
        tree.append(new_edges)
        final_vertices = list({edge[1] for edge in new_edges})
        vertices_seen.extend(final_vertices)
        new_edges = [
            edge
            for edge in edges
            if edge[0] in final_vertices and edge[1] not in
            ↪  vertices_seen and "__init__.py" not in edge[1]
        ]
    return tree
```

This generates the following schema for language model function calling (note: the case below shows a json schema further wrapped in OpenAI's specific "tool" schema):

```
1134
1135   {"tools": [{
1136      "type": "function",
1137      "function": {
1138         "name": "FunctionTestCaseModel",
1139         "description": "Correctly extracted `FunctionTestCaseModel` with
              ↪    all the required parameters with correct types",
1140         "parameters": {
1141            "$defs": {
1142               "ArgsModel": {
1143                  "properties": {
1144                     "module": {
1145                        "description": "Positional argument 'module' with
                             ↪    type '<class 'str'>'",
1146                        "title": "Module",
1147                        "type": "string"
1148                     },
1149                     "edges": {
1150                        "description": "Positional argument 'edges' with
                             ↪    type 'typing.List[typing.Tuple[str, str]]'",
1151                        "items": {
1152                           "items": {"type": "string"},
1153                           "type": "array"
1154                        },
1155                        "title": "Edges",
1156                        "type": "array"
1157                     }
1158                  },
1159                  "required": ["module", "edges"],
1160                  "title": "ArgsModel",
1161                  "type": "object"
1162               },
1163               "KwargsModel": {
1164                  "properties": {},
1165                  "title": "KwargsModel",
1166                  "type": "object"
1167               },
1168               "TestCase": {
1169                  "properties": {
1170                     "args": {
1171                        "allOf": [{"$ref": "#/$defs/ArgsModel"}],
1172                        "description": "Positional args."
1173                     },
1174                     "kwargs": {
1175                        "allOf": [{"$ref": "#/$defs/KwargsModel"}],
1176                        "description": "Keyword args."
1177                     }
1178                  },
1179                  "required": ["args", "kwargs"],
1180                  "title": "TestCase",
1181                  "type": "object"
1182               }
1183            },
1184            "properties": {
1185               "test_cases": {
1186                  "description": "List of test cases",
1187                  "items": {"$ref": "#/$defs/TestCase"},
                     "title": "Test Cases",
                     "type": "array"
                  }
               },
               "required": ["test_cases"],
               "type": "object"
            }
```

```
1188        }
1189  }]]}
1190
1191
1192
1193
```

This schema is then embedded within our test case generation prompt:

```
1194
1195  You are an expert software tester tasked with generating diverse test
1196  ↪   cases for a given function. Your goal is to create a comprehensive
1197  ↪   set of tests that cover various scenarios and edge cases.
1198
1199  First, let's review the previously generated test cases to ensure we
1200  ↪   explore new scenarios:
1201  <previously_generated_test_cases>
1202  {seen or "No test cases have been generated yet."}
1203  </previously_generated_test_cases>
1204
1205  Now, let's examine the context and function details:
1206
1207  <module_code>
1208  {module_code}
1209  </module_code>
1210
1211  Now, let's look at the specific function we need to test:
1212
1213  <function_signature>
1214  {func.signature}
1215  </function_signature>
1216
1217  <function_docstring>
1218  {func.docstring}
1219  </function_docstring>
1220
1221  <function_implementation>
1222  {func.code}
1223  </function_implementation>
1224
1225  Before generating the test cases, let's think through the process:
1226
1227  <test_case_analysis>
1228  1. Analyze the function signature, docstring, and implementation to
1229  ↪   understand its purpose and expected behavior.
1230  2. Identify the input parameters and their types.
1231  3. Determine the function's return type and expected output format.
1232  4. Consider the following categories of test cases:
1233     a. Simple and straightforward cases
1234     b. Complex cases with multiple inputs
1235     c. Edge cases with large values or sizes
1236     d. Edge cases with small values or sizes
1237     e. Cases that may require significant processing time
1238     f. Cases that might cause errors or exceptions
1239     g. Cases with invalid inputs that should raise specific exceptions
1240  5. For numerical arguments:
1241     - Include positive and negative integers/floats
       - Include zero
       - Include prime numbers
       - Include maximum and minimum possible integer values
       - Include very large floats and very small floats (close to zero)
  6. For string arguments:
       - Include empty strings
       - Include strings with special characters
       - Include very long strings
       - Include strings in different languages or with Unicode characters
  7. For boolean arguments:
```

```
    - Include both True and False cases
8. For dynamic containers (e.g., lists, dictionaries):
    - Include cases with many elements
    - Include cases with no elements
    - Include cases with deeply nested objects
    - Include cases with mixed data types
9. For each test case, predict the expected output or exception.
10. Ensure that each test case is unique and covers a different
↪  scenario.
11. Consider any specific constraints or requirements mentioned in the
↪  docstring.
</test_case_analysis>

Now, generate 15 diverse test cases based on this analysis. Present each
↪  test case as a Python dictionary with 'args' and 'kwargs' keys, even
↪  if one of them is empty. Do not include any additional text or
↪  formatting.
```

### A.2.2  TEST GENERATION AND EXECUTION

After generation, each test case is executed against the original function in a controlled environment. We capture:

- Return values or raised exceptions
- Runtime statistics (see Section A.6)

### A.2.3  VALIDATION PIPELINE

Generated test cases are tested against seven validators for quality control. Each validator, upon failure, appends specific feedback as part of a reply to the conversation with the language model:

1. **Schema Conformance:** Test cases must parse as valid function inputs
2. **Duplication:** Each test case input must be unique
3. **Coverage:** Minimum 10 test cases per function
4. **Non-triviality:** Less than 50% of cases can return unmodified input
5. **Output Diversity:** No single output as 66% of cases
6. **Error Balance:** Exception cases limited to 50% of total
7. **Runtime Bounds:** CPU time under 10 seconds per case

We provide examples of validation feedback messages in Section A.4.

### A.2.4  REGENERATION STRATEGY

The system allows two full generation attempts, each permitting three validation/reply/regeneration cycles. To maximise task breadth while maintaining quality, we may still preserve some test cases from a task that fails to pass all validators. We do this by relaxing some validator requirements:

- The minimum test case count requirement (criterion 3) is waived for the final generation attempt
- Test cases that contain duplicates or exceed runtime bounds are individually filtered out (criteria 2 and 7)
- The task's remaining test cases must still meet our core quality requirements: non-triviality, output diversity, and a balanced error rate (criteria 4, 5, and 6)

This approach using GPT-4o-latest (generation spanned multiple versions) yields our current dataset of 33,875 test cases across 1,000 repositories, with an average generation cost of 1,123 tokens per

test case. Failed generations primarily occur due to schema conformance (criterion 1 - schema conformance poses an outsized challenge to smaller models i.e. llama3.1-8b; mirroring the execution prediction task), duplication and output diversity (criterion 2 and 5 - both commonly observed in functions with a limited input/output domains, i.e. single boolean args/ returns).

## A.3 FUNCTION SELECTION AND DEPENDENCY RESOLUTION

The function selection and dependency collation process comprises three main stages: type annotation analysis, dependency graph construction, and code inlining, followed by a final filtering stage. Here we detail each stage:

### A.3.1 TYPE ANNOTATION ANALYSIS

Function selection begins with a recursive AST analysis of type annotations. Each candidate function must have both its arguments and return type validated as "LLM-generatable" - meaning they can be reliably produced by a language model. As detailed in Section A.5, we recursively validate against a predefined set of acceptable types.

For example, when processing complex nested types like 'List[Tuple[str, int]]', the analyzer first validates 'List', then 'Tuple', and finally 'str' and 'int'. Functions with arguments or return types containing non-LLM-generatable elements (e.g., file handles, sockets, custom objects) are filtered out during this stage.

### A.3.2 DEPENDENCY GRAPH CONSTRUCTION

Once a function passes type validation, we construct a dependency graph to identify all code elements required for the function's execution. This process involves:

1. **Symbol Analysis:** For each function, we perform an AST walk to identify:

- Local variables: We track symbols defined within the current scope including but not limited to assignments, function arguments, loop variables, comprehension variables, and lambda parameters. These are excluded from dependency tracking as they are part of the function's internal logic.

- Used symbols: We collect all variable references, function calls, type annotations (e.g., in 'x: List[Prompt]', both 'List' and 'Prompt' need resolving), and attribute accesses (e.g., in 'library.varname', both 'library' and 'varname' need resolving). By comparing against the local variables, we identify which symbols must be resolved externally. For each symbol, we walk the AST to find its definition.

- Nested scopes: We handle nested functions and classes by treating their names as local variables in the outer scope while tracking their internal symbol usage separately.

2. **Import Resolution:** For each identified external dependency, we:

- Resolve relative imports based on the file's location in the package and module imports based on the package structure

- Track aliases and renamed imports, mapping against accessed attributes (e.g. for 'lib.var' where we 'import x as lib', we must find 'var' in 'x')

- Ignore builtin imports, treating them as standard code blocks

- Recursively process imported modules, classes, functions and variables through Step 1. Symbol Analysis

- Handle special cases such as '\_\_init\_\_.py' files, complex imports 'from x import *' and more

3. **Graph Construction:** We build a directed graph where nodes represent code blocks (functions, classes, assignments) and edges represent dependencies between these blocks. The graph maintains the minimal set of dependencies required for each function while preserving their original relationships.

4. **Symbol Resolution Validation:** Before a function is accepted, we verify that every used symbol has been successfully resolved to a definition. This validation is crucial as it ensures we can create a complete, self-contained version of the function. Functions using runtime code generation (e.g., 'exec', 'eval'), dynamic attribute access (e.g., 'getattr' with variable names), or other patterns that prevent static resolution are largely filtered out at this stage.

To illustrate this process with a simple example, consider the following from the Azure SDK Tables package. The original code was spread across two files in 'azure-nspkg/sdk/tables/azure-data-tables/azure/data/tables/'. The extracted minimal dependency chain (debug output preserved to show file origins and dependency types) is shown below:

```python
# _common_conversion.py | resolved_import_from/defaultlib -> from
↪   datetime import timezone
from datetime import timezone

# _common_conversion.py | function -> _to_utc_datetime
def _to_utc_datetime(value):
    try:
        value = value.astimezone(timezone.utc)
    except ValueError:
        # Before Python 3.8, this raised for a naive datetime.
        pass
    try:
        return value.strftime("%Y-%m-%dT%H:%M:%S.%fZ")
    except ValueError:
        return value.strftime("%Y-%m-%dT%H:%M:%SZ")

# _serialize.py | resolved_import_from/defaultlib -> from datetime
↪   import datetime
from datetime import datetime
# _serialize.py | resolved_import_from/defaultlib -> from uuid import
↪   UUID
from uuid import UUID
# _serialize.py | resolved_import_from/defaultlib -> from typing import
↪   Dict, Optional, Union
from typing import Dict, Optional
# _serialize.py | resolved_import_from/defaultlib -> from binascii
↪   import hexlify
from binascii import hexlify

# _serialize.py | function -> _parameter_filter_substitution
def _parameter_filter_substitution(parameters: Optional[Dict[str, str]],
↪   query_filter: str) -> str:
    """Replace user defined parameters in filter.
    :param parameters: User defined parameters
    :type parameters: dict[str, str]
    :param str query_filter: Filter for querying
    :return: A query filter replaced by user defined parameters.
    :rtype: str
    """
    if parameters:
        filter_strings = query_filter.split(" ")
        for index, word in enumerate(filter_strings):
            if word[0] == "@":
                val = parameters[word[1:]]
                if val in [True, False]:
                    filter_strings[index] = str(val).lower()
                elif isinstance(val, (float)):
                    filter_strings[index] = str(val)
                elif isinstance(val, int):
                    if val.bit_length() <= 32:
                        filter_strings[index] = str(val)
                    else:
                        filter_strings[index] = f"{str(val)}L"
```

```
1404
1405            elif isinstance(val, datetime):
1406                filter_strings[index] =
                ↪ f"datetime'{_to_utc_datetime(val)}'"
1407            elif isinstance(val, UUID):
1408                filter_strings[index] = f"guid'{str(val)}'"
1409            elif isinstance(val, bytes):
1410                v = str(hexlify(val))
1411                v = v[2:-1]  # Python 3 adds a 'b' and quotations
1412                filter_strings[index] = f"X'{v}'"
1413            else:
1414                val = val.replace("'", "''")
1415                filter_strings[index] = f"'{val}'"
1416        return " ".join(filter_strings)
        return query_filter
```

Note that these functions have been extracted from much larger source files (indicated by the commented file names) - we only collect the minimal code required for execution.

### A.3.3 CODE INLINING

The final stage involves generating a self-contained version of the function with all dependencies inlined. Rather than attempting to strip back the original files to their minimal form, we are motivated to inline as it ensures the language model executes exactly the same code as our interpreter.

The inlining process:

1. Performs a topological sort of the dependency graph to determine the correct order of declarations.

2. Inlines code based on its original structure:

- Most code, including functions, classes, and variables, is inlined directly at the appropriate scope.
- When an entire module has been imported (e.g., 'import random'), we create a namespace class to maintain proper module-level scoping.

3. Generates the final code by maintaining the original code structure and ensuring all dependencies are declared before use.

After code inlining, we perform a final filtering pass to remove functions with side effects or non-deterministic behaviour. This filtering must occur after inlining as many problematic patterns only become apparent once we have the complete code context. For example, network requests might be hidden behind multiple layers of function calls, or environment variables might be accessed through utility functions in separate modules. Functions that use system calls, file I/O, network operations, random number generation, or environment variables are filtered out at this stage.

While our dependency resolution system handles many common Python patterns, including dynamic imports and aliased imports, there remain some challenges. Functions with circular dependencies between modules cannot currently be processed, and certain package initialization patterns that rely on import-time side effects are not supported. These limitations primarily affect a small percentage of candidate functions.

### A.4 VALIDATOR EXAMPLES

Each validator appends specific feedback to guide the model in correcting errors. Below are the prompts used for each of these feedback messages:

### A.4.1 SCHEMA CONFORMANCE VALIDATOR

```
Validation Error found while parsing test case JSON:
    <exception>
```

```
{exception}
</exception>
Recall the function correctly, fix these errors and generate a valid
↪  test case following the schema.
```

### A.4.2  DUPLICATION VALIDATOR

```
Validation Error: Duplicate test case inputs detected
    The following test cases have identical inputs:
    <duplicate_cases>
    {json.dumps(duplicate_cases)}
    </duplicate_cases>
    Recall the function correctly and generate test cases with unique
    ↪  input combinations.
```

### A.4.3  COVERAGE VALIDATOR

```
Validation Error: Insufficient test coverage ({len(cases)}/10 required
↪  minimum cases).
    Generate additional unqiue test cases to cover these scenarios.
```

### A.4.4  NON-TRIVIALITY VALIDATOR

```
Validation Error: Test cases too simple. Greater than 50% of test cases
↪  are returning their inputs as outputs. Inputs must undergo some
↪  transformation during processing.
    <test_cases_with_results>
    {json.dumps(cases)}
    </test_cases_with_results>
    Fix these errors by generating test cases that:
    1. Explore different code paths within the function
    2. Trigger transformation of the inputs so that they differ from the
    ↪  outputs
```

### A.4.5  OUTPUT DIVERSITY VALIDATOR

```
Validation Error: Insufficient output diversity in test cases. One
↪  output is returned by more than 2/3s of all cases.
    <test_cases_counted_outputs>
    {json.dumps(output_counter)}
    <test_cases_counted_outputs>
    <test_cases_with_results>
    {json.dumps(cases)}
    </test_cases_with_results>
    Generate additional test cases that contain differing outputs to the
    ↪  most popular above.
```

### A.4.6  ERROR BALANCE VALIDATOR

```
Validation Error: Too many error-inducing test cases
↪  ({len(error_cases)}/{len(cases)})
    <test_cases_with_results>
    {json.dumps(cases)}
    </test_cases_with_results>
```

## A.5   ACCEPTABLE TYPES & FILTERING CRITERIA

**Acceptable types.** To find functions where the inputs and outputs are LLM generatable, we recursively parse both arguments and return types as AST objects i.e. for `list[tuple[str, False]]` we first check `list` is an acceptable type, then recurse down into `tuple`, following that we then check `str` and finally we check `False`. `False` isn't an acceptable type but it is an acceptable constant and hence accepted. Note: certain acceptable types and constants are not allowed as return values, i.e. `None` is not an accepted return constant

acceptable_types = { 'int', 'str', 'float', 'bool', 'none', 'list', 'dict', 'tuple', 'set', 'datetime.date', 'date', 'literal', 'optional', 'union', 'sequence', 'iterable', 'frozenset', 'mapping' }

acceptable_constants = { 'ellipsis', True, False, None }

**Filtering functions.** When filtering functions we maintain four separate block lists, 1) a list of banned imports (including direct and aliases), 2) a list of banned functions (some common libraries have a limited set of non-deterministic methods, we don't want to fully exclude them), 3) a list of banned variables (some variables such as `__version__` are likely to be environment based), 4) a list of banned repos (some repos from cloud providers provide thousands of near identical methods with different urls, we remove these as they are not a valuable contribution to the evaluation).

## A.6   STATIC AND RUNTIME CODE STATISTICS

Given a task from the evaluation set we perform the following static and runtime analyses:

**Static Analysis:**

1. **Lines of Code Count**. Total number of lines, excluding blanks and comments.
2. **AST Node Types Count**. Count of all Python Abstract Syntax Tree (AST) node types present in the code, e.g. `FunctionDef()`, `AsyncFunctionDef()`, `Assign()`, `For()`, ...
3. **Cyclomatic Complexity**. An estimate of the number of linearly independent paths through a program's source code. Note: There are several limitations in the implementation of this metric as we only parse python source code, and some modern python features such as pattern matching statements are yet to be supported.
4. **Maintainability Index**. A estimate of code maintainability and quality incorporating several other estimated measures (e.g. Halstead Volume, Cyclomatic Complexity, and lines of code). Note: Faces the same aforementioned limitations.

**Runtime Analysis:**

1. **CPU Time.**
2. **Loop Iterations**. Including for loops, while loops and list comprehensions.
3. **Arithmetic Operations**. Including addition, subtraction, multiplication, division and power operations.
4. **Execution Metrics**. Including lines executed, library lines executed and conditional statements executed.
5. **Function Calls**. Including builtin function calls, user-defined function calls and total function calls.
6. **Variable Usage**. Including variables declared and variables used

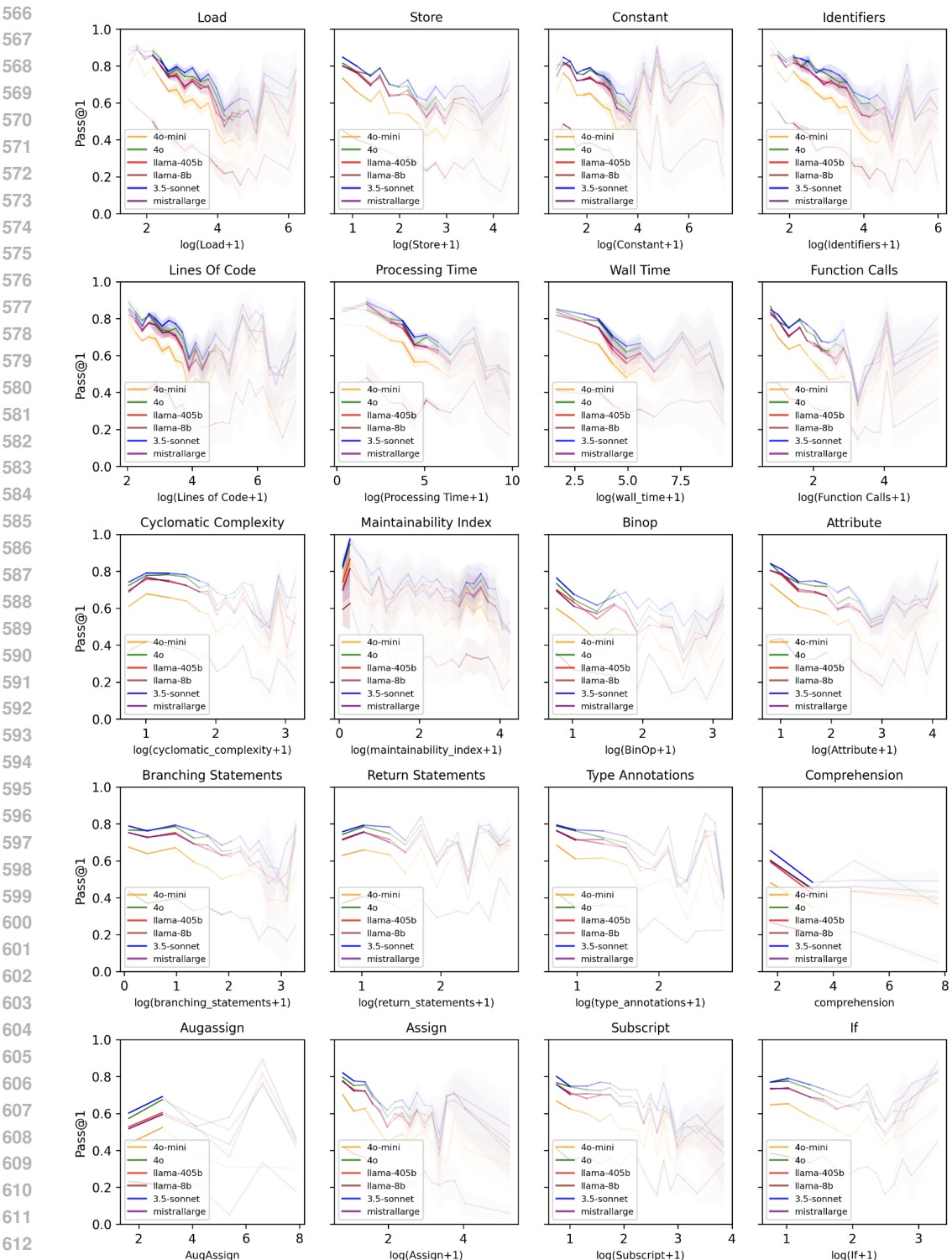

Figure 9: Top static statistics visualised against Pass@1 rate for all models tested

## A.7 OUTPUT COMPARISON AND VALIDATION

When evaluating model outputs against ground truth values, we employ two distinct comparison strategies depending on whether the output represents a successful execution result or an error case.

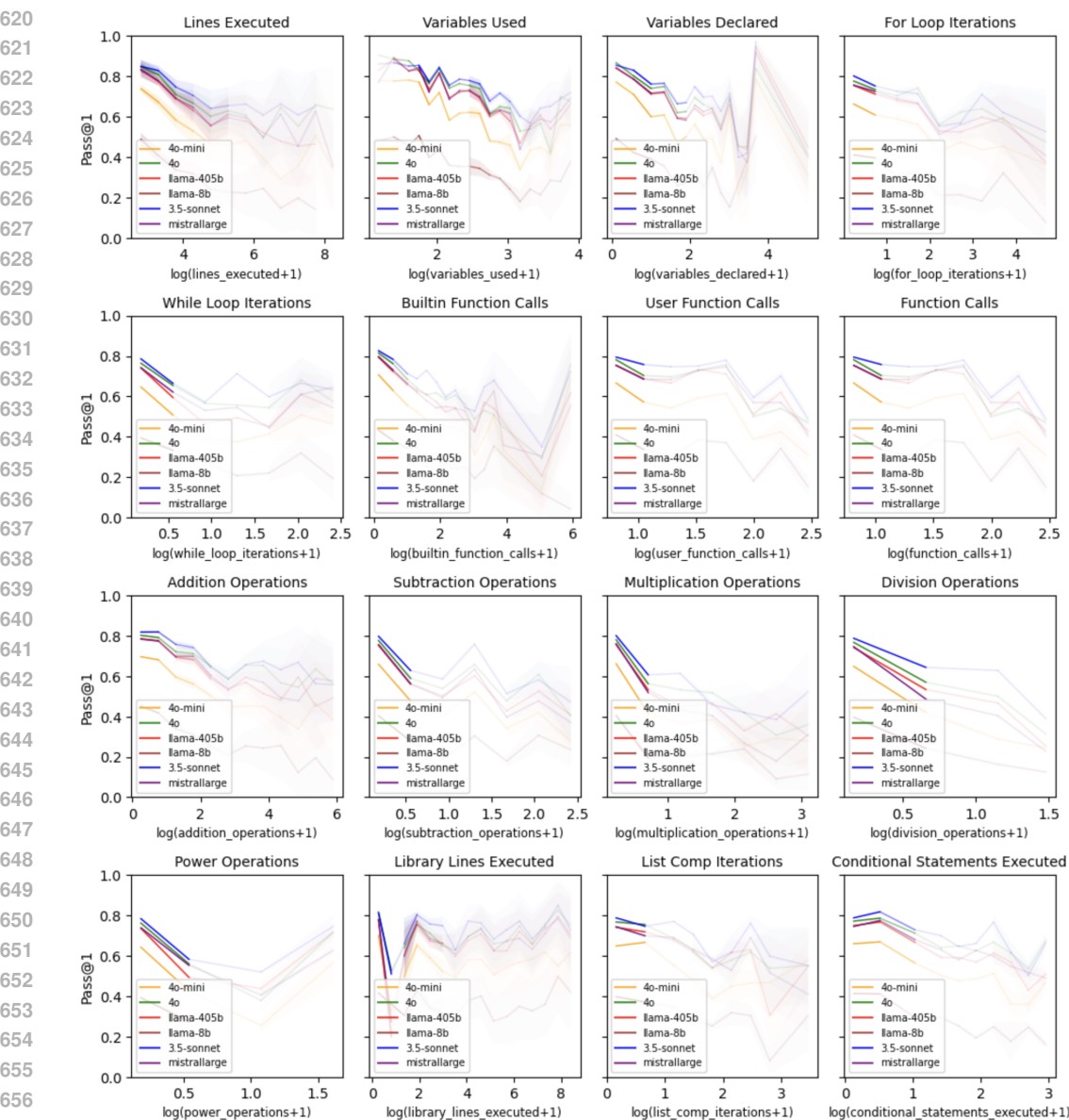

Figure 10: Runtime statistics visualised against Pass@1 rate for all models tested

This dual approach is necessary because error messages often contain version or implementation-specific details while maintaining semantic equivalence.

### A.7.1 DIRECT VALUE COMPARISON

For successful execution results, we perform limited preprocessing (unsorted container objects e.g. `set` and `frozenset` are sorted before conversion to json lists, iterable types i.e. tuples are converted to lists, numbers are consistently formatted), then make a direct comparison between the model output and ground truth as json objects.

### A.7.2 ERROR MESSAGE COMPARISON

For error cases, we use a language model-based comparison approach that focuses on specific error patterns and known version differences. This structured approach is necessary as error messages have evolved across Python versions while maintaining the same underlying causes.

**Stacktrace Handling.** We explicitly exclude stacktraces from comparison as they contain execution-specific information like file paths, and external details that the model is not privy to.

**Version-Specific Error Messages.** Python has evolved to provide more helpful error messages in recent versions, with significant changes between major releases. Our comparison system must handle these variations appropriately. Examples of version-specific differences:

```python
# Python 3.9
my_list = [1, 2 3]
SyntaxError: invalid syntax

# Python 3.10
my_list = [1, 2 3]
SyntaxError: invalid syntax. Perhaps you forgot a comma?

# Both indicate the same missing comma issue

# Python 3.11
my_string = f"{x z y}" + f"{1 + 1}"
SyntaxError: f-string: invalid syntax. Perhaps you forgot a comma?

# Python 3.12
my_string = f"{x z y}" + f"{1 + 1}"
SyntaxError: invalid syntax. Perhaps you forgot a comma?

# While the messages differ, they point to the same syntactic error
```

To handle these variations, our error comparison system uses a prompt that encourages human-like reasoning about error equivalence:

```
You are an expert Python developer looking at two error messages.
↪  Determine if they are describing the same underlying issue, even if
↪  expressed differently. Consider:

- Different Python versions might provide different levels of detail for
↪  the same error
- The core issue might be described in more or less helpful ways
- Extra hints or suggestions don't change the fundamental error
- Line numbers and file paths are irrelevant

Message 1: {error1}
Message 2: {error2}

Would a Python developer consider these to be the same error? Answer
↪  only 'True' or 'False'.
```

This structured approach to error comparison improves consistency in evaluation across different Python versions and implementation variations while maintaining the ability to identify truly distinct error cases.

## A.8 PER FUNCTION TASK SET DIVERSITY

To measure EXE's potential to scale in the future, we analyse a model's ability to continually generate new test cases given a single function. This is performed by:

1. Sampling functions from EXE's dataset (samples detailed below).
2. Generating a batch of test cases in accordance with A.2, recording token usage.

3. Running validators in accordance with A.2, removing cases that are duplicates, fail to execute, fail to be parsed, or that trigger any validator.

4. Continue generating new batches of test cases, injecting a random selection of (up to 60) previously generated cases into the prompt (detailed samples of test cases generated can be seen at the end of this appendix).

### A.8.1 INLINED CODE TASKS FOR GENERATION

Example 1. `get_origin_link_and_tag` from `utils.py` in azure-nspkg:

```python
from typing import List

def get_origin_link_and_tag(issue_body_list: List[str]) -> (str, str):
    link, readme_tag = '', ''
    for row in issue_body_list:
        if 'link' in row.lower() and 'release request' not in
        ↪  row.lower() and link == '':
            link = row.split(":", 1)[-1].strip()
        if 'readme tag' in row.lower() and readme_tag == '':
            readme_tag = row.split(":", 1)[-1].strip()
        if link and readme_tag:
            break

    if link.count('https') > 1:
        link = link.split(']')[0]
        link = link.replace('[', "").replace(']', "").replace('(',
        ↪  "").replace(')', "")
    return link, readme_tag
```

Example 2. `_compute_affine_output_size_python.py` from `geometry.py` in torchvision:

```python
from typing import List, Tuple

import math

def _compute_affine_output_size_python(matrix: List[float], w: int, h:
↪  int) -> Tuple[int, int]:
    # Mostly copied from PIL implementation:
    # The only difference is with transformed points as input matrix has
    ↪   zero translation part here and
    # PIL has a centered translation part.
    # https://github.com/python-pillow/Pillow/blob/11de3318867e43980573⌋
    ↪   73ee9f12dcb33db7335c/src/PIL/Image.py#L2054

    a, b, c, d, e, f = matrix
    xx = []
    yy = []

    half_w = 0.5 * w
    half_h = 0.5 * h
    for x, y in ((-half_w, -half_h), (half_w, -half_h), (half_w,
    ↪  half_h), (-half_w, half_h)):
        nx = a * x + b * y + c
        ny = d * x + e * y + f
        xx.append(nx + half_w)
        yy.append(ny + half_h)

    nw = math.ceil(max(xx)) - math.floor(min(xx))
    nh = math.ceil(max(yy)) - math.floor(min(yy))
    return int(nw), int(nh)  # w, h
```

Example 3. `_format_image.py` from `_chat_models.py` in langchain-core:

```python
from typing import Dict

import re

def _format_image(image_url: str) -> Dict:
    """
    Formats an image of format data:image/jpeg;base64,{b64_string}
    to a dict for anthropic api

    {
      "type": "base64",
      "media_type": "image/jpeg",
      "data": "/9j/4AAQSkZJRg...",
    }

    And throws an error if it's not a b64 image
    """
    regex = r"^data:(?P<media_type>image/.+);base64,(?P<data>.+)$"
    match = re.match(regex, image_url)
    if match is None:
        raise ValueError(
            "Anthropic only supports base64-encoded images currently."
            " Example: data:image/png;base64,'/9j/4AAQSk'..."
        )
    return {
        "type": "base64",
        "media_type": match.group("media_type"),
        "data": match.group("data"),
    }
```

Example 4. `make_arn_for_alarm.py` from `utils.py` in moto:

```python
REGION_PREFIX_TO_PARTITION = {
    # (region prefix, aws partition)
    "cn-": "aws-cn",
    "us-gov-": "aws-us-gov",
    "us-iso-": "aws-iso",
    "us-isob-": "aws-iso-b",
}

DEFAULT_PARTITION = "aws"

PARTITION_NAMES = list(REGION_PREFIX_TO_PARTITION.values()) +
↪    [DEFAULT_PARTITION]

def get_partition(region: str) -> str:
    if not region:
        return DEFAULT_PARTITION
    if region in PARTITION_NAMES:
        return region
    for prefix in REGION_PREFIX_TO_PARTITION:
        if region.startswith(prefix):
            return REGION_PREFIX_TO_PARTITION[prefix]
    return DEFAULT_PARTITION

def make_arn_for_alarm(region: str, account_id: str, alarm_name: str) ->
↪    str:
    return
    ↪    f"arn:{get_partition(region)}:cloudwatch:{region}:{account_id}:alarm:{alarm_name}"
```

Example 5. `number2lowercase_roman_numeral.py` from `page_labels.py` in pypdf2:

```python
from typing import Iterator

def number2uppercase_roman_numeral(num: int) -> str:
    roman = [
        (1000, "M"),
        (900, "CM"),
        (500, "D"),
        (400, "CD"),
        (100, "C"),
        (90, "XC"),
        (50, "L"),
        (40, "XL"),
        (10, "X"),
        (9, "IX"),
        (5, "V"),
        (4, "IV"),
        (1, "I"),
    ]

    def roman_num(num: int) -> Iterator[str]:
        for decimal, roman_repr in roman:
            x, _ = divmod(num, decimal)
            yield roman_repr * x
            num -= decimal * x
            if num <= 0:
                break

    return "".join(list(roman_num(num)))

def number2lowercase_roman_numeral(number: int) -> str:
    return number2uppercase_roman_numeral(number).lower()
```

Example 6. `alpha_canonicalize.py` from `parser.py` in opt-einsum:

```python
_einsum_symbols_base = \
    "abcdefghijklmnopqrstuvwxyzABCDEFGHIJKLMNOPQRSTUVWXYZ"

from typing import Dict

def get_symbol(i: int) -> str:
    """Get the symbol corresponding to int ``i`` - runs through the
    usual 52
    letters before resorting to unicode characters, starting at
    ``chr(192)`` and skipping surrogates.

    **Examples:**

    ```python
    get_symbol(2)
    #> 'c'

    get_symbol(200)
    #> 'Ŕ'

    get_symbol(20000)
    #> '',
    ```
    """
    if i < 52:
        return _einsum_symbols_base[i]
```

```python
    elif i >= 55296:
        # Skip chr(57343) - chr(55296) as surrogates
        return chr(i + 2048)
    else:
        return chr(i + 140)

def alpha_canonicalize(equation: str) -> str:
    """Alpha convert an equation in an order-independent canonical way.

    Examples
    --------
    >>> oe.parser.alpha_canonicalize("dcba")
    'abcd'

    >>> oe.parser.alpha_canonicalize("Ĥěḻḻö")
    'abccd'
    """
    rename: Dict[str, str] = {}
    for name in equation:
        if name in ".,->":
            continue
        if name not in rename:
            rename[name] = get_symbol(len(rename))
    return "".join(rename.get(x, x) for x in equation)
```

Example 7. `remove_starting_comments.py` from `sql_util.py` in snowflake-connector-python:

```python
import re

COMMENT_START_SQL_RE = re.compile(
    r"""
                        ^\s*(?:
                            /\*[\w\W]*?\*/
                        )""",
    re.VERBOSE,
)

def remove_starting_comments(sql: str) -> str:
    """Remove all comments from the start of a SQL statement."""
    commentless_sql = sql
    while True:
        start_comment = COMMENT_START_SQL_RE.match(commentless_sql)
        if start_comment is None:
            break
        commentless_sql = commentless_sql[start_comment.end() :]
    return commentless_sql
```

Example 8. `_pad_version.py` from `specifiers.py` in poetry-core:

```python
import itertools

from typing import List, Tuple

def _pad_version(left: List[str], right: List[str]) -> Tuple[List[str],
↪  List[str]]:
    left_split, right_split = [], []

    # Get the release segment of our versions
    left_split.append(list(itertools.takewhile(lambda x: x.isdigit(),
    ↪  left)))
```

```
1944
1945       right_split.append(list(itertools.takewhile(lambda x: x.isdigit(),
1946    ↪   right)))

1947       # Get the rest of our versions
1948       left_split.append(left[len(left_split[0]) :])
1949       right_split.append(right[len(right_split[0]) :])

1950
1951       # Insert our padding
1952       left_split.insert(1, ["0"] * max(0, len(right_split[0]) -
          ↪   len(left_split[0])))
1953       right_split.insert(1, ["0"] * max(0, len(left_split[0]) -
1954    ↪   len(right_split[0])))

1955
1956       return (list(itertools.chain(*left_split)),
          ↪   list(itertools.chain(*right_split)))
1957
1958
```

Example 9. `get_flag_suggestions.py` from `_helpers.py` in absl-py:

```
1961    _SUGGESTION_ERROR_RATE_THRESHOLD = 0.50

1962
1963    from typing import List, Sequence

1964    def _damerau_levenshtein(a, b):
1965      """Returns Damerau-Levenshtein edit distance from a to b."""
1966      memo = {}

1967
1968      def distance(x, y):
1969        """Recursively defined string distance with memoization."""
1970        if (x, y) in memo:
          return memo[x, y]
1971        if not x:
1972          d = len(y)
1973        elif not y:
          d = len(x)
1974        else:
1975          d = min(
1976              distance(x[1:], y) + 1,   # correct an insertion error
              distance(x, y[1:]) + 1,   # correct a deletion error
1977              distance(x[1:], y[1:]) + (x[0] != y[0]))   # correct a wrong
              ↪   character
1978          if len(x) >= 2 and len(y) >= 2 and x[0] == y[1] and x[1] == y[0]:
1979            # Correct a transposition.
1980            t = distance(x[2:], y[2:]) + 1
1981            if d > t:
              d = t
1982
1983
1984        memo[x, y] = d
1985        return d
1986      return distance(a, b)

1987
1988    def get_flag_suggestions(
1989        attempt: str, longopt_list: Sequence[str]
        ) -> List[str]:
1990      """Returns helpful similar matches for an invalid flag."""
1991      # Don't suggest on very short strings, or if no longopts are
        ↪   specified.
1992      if len(attempt) <= 2 or not longopt_list:
1993        return []

1994
1995      option_names = [v.split('=')[0] for v in longopt_list]

1996
1997      # Find close approximations in flag prefixes.
        # This also handles the case where the flag is spelled right but
        ↪   ambiguous.
```

```
distances = [(_damerau_levenshtein(attempt, option[0:len(attempt)]),
↪ option)
              for option in option_names]
# t[0] is distance, and sorting by t[1] allows us to have stable
↪ output.
distances.sort()

least_errors, _ = distances[0]
# Don't suggest excessively bad matches.
if least_errors >= _SUGGESTION_ERROR_RATE_THRESHOLD * len(attempt):
  return []

suggestions = []
for errors, name in distances:
  if errors == least_errors:
    suggestions.append(name)
  else:
    break
return suggestions
```

Example 10. `valid_contexto.py` from `core.py` in idna:

```python
from types import SimpleNamespace

from typing import Tuple

def _encode_range(start: int, end: int) -> int:
    return (start << 32) | end

def _decode_range(r: int) -> Tuple[int, int]:
    return (r >> 32), (r & ((1 << 32) - 1))

import bisect

def intranges_contain(int_: int, ranges: Tuple[int, ...]) -> bool:
    """Determine if `int_` falls into one of the ranges in `ranges`."""
    tuple_ = _encode_range(int_, 0)
    pos = bisect.bisect_left(ranges, tuple_)
    # we could be immediately ahead of a tuple (start, end)
    # with start < int_ <= end
    if pos > 0:
        left, right = _decode_range(ranges[pos-1])
        if left <= int_ < right:
            return True
    # or we could be immediately behind a tuple (int_, end)
    if pos < len(ranges):
        left, _ = _decode_range(ranges[pos])
        if left == int_:
            return True
    return False

class idnadataClass(SimpleNamespace):
    def __init__(self):
        scripts = {
            'Greek': (
                0x37000000374,
                0x37500000378,
                0x37a0000037e,
                0x37f00000380,
                0x38400000385,
                0x38600000387,
                0x3880000038b,
                0x38c0000038d,
```

```
                  0x38e000003a2,
                  0x3a3000003e2,
                  0x3f000000400,
                  0x1d2600001d2b,
                  0x1d5d00001d62,
                  0x1d6600001d6b,
                  0x1dbf00001dc0,
                  0x1f0000001f16,
                  0x1f1800001f1e,
                  0x1f2000001f46,
                  0x1f4800001f4e,
                  0x1f5000001f58,
                  0x1f5900001f5a,
                  0x1f5b00001f5c,
                  0x1f5d00001f5e,
                  0x1f5f00001f7e,
                  0x1f8000001fb5,
                  0x1fb600001fc5,
                  0x1fc600001fd4,
                  0x1fd600001fdc,
                  0x1fdd00001ff0,
                  0x1ff200001ff5,
                  0x1ff600001fff,
                  0x212600002127,
                  0xab650000ab66,
                  0x101400001018f,
                  0x101a0000101a1,
                  0x1d2000001d246,
              ),
              'Han': (
                  0x2e8000002e9a,
                  0x2e9b00002ef4,
                  0x2f0000002fd6,
                  0x300500003006,
                  0x300700003008,
                  0x30210000302a,
                  0x30380000303c,
                  0x340000004dc0,
                  0x4e000000a000,
                  0xf9000000fa6e,
                  0xfa700000fada,
                  0x16fe200016fe4,
                  0x16ff000016ff2,
                  0x200000002a6e0,
                  0x2a7000002b73a,
                  0x2b7400002b81e,
                  0x2b8200002cea2,
                  0x2ceb00002ebe1,
                  0x2ebf00002ee5e,
                  0x2f8000002fa1e,
                  0x300000003134b,
                  0x31350000323b0,
              ),
              'Hebrew': (
                  0x591000005c8,
                  0x5d0000005eb,
                  0x5ef000005f5,
                  0xfb1d0000fb37,
                  0xfb380000fb3d,
                  0xfb3e0000fb3f,
                  0xfb400000fb42,
                  0xfb430000fb45,
                  0xfb460000fb50,
              ),
```

```
2106
2107                'Hiragana': (
2108                    0x304100003097,
2109                    0x309d000030a0,
2110                    0x1b0010001b120,
2111                    0x1b1320001b133,
2112                    0x1b1500001b153,
                       0x1f2000001f201,
2113                ),
2114                'Katakana': (
2115                    0x30a1000030fb,
2116                    0x30fd00003100,
2117                    0x31f000003200,
2118                    0x32d0000032ff,
                       0x330000003358,
2119                    0xff660000ff70,
2120                    0xff710000ff9e,
2121                    0x1aff00001aff4,
2122                    0x1aff50001affc,
                       0x1affd0001afff,
2123                    0x1b0000001b001,
2124                    0x1b1200001b123,
2125                    0x1b1550001b156,
                       0x1b1640001b168,
2126                ),
2127            }

2128
2129            self.__dict__.update(locals())

2130    idnadata = idnadataClass()

2131
2132    def _is_script(cp: str, script: str) -> bool:
2133        return intranges_contain(ord(cp), idnadata.scripts[script])

2134    def valid_contexto(label: str, pos: int, exception: bool = False) ->
2135    ↪    bool:
2136        cp_value = ord(label[pos])

2137
2138        if cp_value == 0x00b7:
            if 0 < pos < len(label)-1:
2139                if ord(label[pos - 1]) == 0x006c and ord(label[pos + 1]) ==
2140                ↪    0x006c:
2141                    return True
2142            return False

2143
2144        elif cp_value == 0x0375:
            if pos < len(label)-1 and len(label) > 1:
2145                return _is_script(label[pos + 1], 'Greek')
2146            return False

2147
2148        elif cp_value == 0x05f3 or cp_value == 0x05f4:
2149            if pos > 0:
2150                return _is_script(label[pos - 1], 'Hebrew')
            return False

2151
2152        elif cp_value == 0x30fb:
2153            for cp in label:
                if cp == '\u30fb':
2154                    continue
2155                if _is_script(cp, 'Hiragana') or _is_script(cp, 'Katakana')
2156                ↪    or _is_script(cp, 'Han'):
2157                    return True
2158            return False

2159        elif 0x660 <= cp_value <= 0x669:
```

```
2160
2161         for cp in label:
2162             if 0x6f0 <= ord(cp) <= 0x06f9:
                     return False
2163         return True
2164
2165     elif 0x6f0 <= cp_value <= 0x6f9:
2166         for cp in label:
2167             if 0x660 <= ord(cp) <= 0x0669:
                     return False
2168         return True
2169
2170     return False
2171
2172
```

Example 11. `exact_match.py` from `meteor_score.py` in nltk:

```
2174  from typing import Callable, Iterable, List, Tuple
2175
2176  def _match_enums(
2177      enum_hypothesis_list: List[Tuple[int, str]],
2178      enum_reference_list: List[Tuple[int, str]],
      ) -> Tuple[List[Tuple[int, int]], List[Tuple[int, str]], List[Tuple[int,
2179  ↪   str]]]:
2180      """
2181      matches exact words in hypothesis and reference and returns
2182      a word mapping between enum_hypothesis_list and enum_reference_list
2183      based on the enumerated word id.
2184      :param enum_hypothesis_list: enumerated hypothesis list
2185      :param enum_reference_list: enumerated reference list
2186      :return: enumerated matched tuples, enumerated unmatched hypothesis
         ↪   tuples,
2187              enumerated unmatched reference tuples
2188      """
2189      word_match = []
2190      for i in range(len(enum_hypothesis_list))[::-1]:
2191          for j in range(len(enum_reference_list))[::-1]:
2192              if enum_hypothesis_list[i][1] == enum_reference_list[j][1]:
2193                  word_match.append(
2194                      (enum_hypothesis_list[i][0],
                         ↪   enum_reference_list[j][0])
2195                  )
2196                  enum_hypothesis_list.pop(i)
2197                  enum_reference_list.pop(j)
2198                  break
      return word_match, enum_hypothesis_list, enum_reference_list
2199
2200  def _generate_enums(
2201      hypothesis: Iterable[str],
2202      reference: Iterable[str],
2203      preprocess: Callable[[str], str] = str.lower,
2204  ) -> Tuple[List[Tuple[int, str]], List[Tuple[int, str]]]:
      """
2205      Takes in pre-tokenized inputs for hypothesis and reference and
         ↪   returns
2206      enumerated word lists for each of them
2207
2208      :param hypothesis: pre-tokenized hypothesis
2209      :param reference: pre-tokenized reference
2210      :preprocess: preprocessing method (default str.lower)
2211      :return: enumerated words list
2212      """
2213      if isinstance(hypothesis, str):
          raise TypeError(
```

```
2214
2215              f'"hypothesis" expects pre-tokenized hypothesis
2216          ↪  (Iterable[str]): {hypothesis}'
2217      )
2218
      if isinstance(reference, str):
2219          raise TypeError(
2220              f'"reference" expects pre-tokenized reference
2221          ↪  (Iterable[str]): {reference}'
2222          )
2223
      enum_hypothesis_list = list(enumerate(map(preprocess, hypothesis)))
2224      enum_reference_list = list(enumerate(map(preprocess, reference)))
2225      return enum_hypothesis_list, enum_reference_list
2226
  def exact_match(
2227      hypothesis: Iterable[str], reference: Iterable[str]
2228  ) -> Tuple[List[Tuple[int, int]], List[Tuple[int, str]], List[Tuple[int,
2229  ↪  str]]]:
2230      """
2231      matches exact words in hypothesis and reference
      and returns a word mapping based on the enumerated
2232      word id between hypothesis and reference
2233
2234      :param hypothesis: pre-tokenized hypothesis
2235      :param reference: pre-tokenized reference
      :return: enumerated matched tuples, enumerated unmatched hypothesis
2236      ↪  tuples,
2237              enumerated unmatched reference tuples
2238      """
2239      enum_hypothesis_list, enum_reference_list =
2240      ↪  _generate_enums(hypothesis, reference)
      return _match_enums(enum_hypothesis_list, enum_reference_list)
2241
2242
2243
```

A.8.2 GENERATED TEST CASES

Below is a sample of generated test cases (cut down to 3 examples, showing the first 60 cases for brevity).

First 60 generated cases for `alpha_canonicalize`

```
0:   {"args": ["a1b2c3->d4e5f6"], "kwargs": {}}
1:   {"args": ["    ->    ,    ->    "], "kwargs": {}}
2:   {"args": ["   "], "kwargs": {}}
3:   {"args": ["AAA BBB CCC"], "kwargs": {}}
4:   {"args": ["abcdefghijklmnopqrstuvwxyz"], "kwargs": {}}
5:   {"args": ["a\u0000b\u0001c\u0002->d\u0003e\u0004f\u0005"],
↪  "kwargs": {}}
6:   {"args": ["abcdefghijklmnopqrstuvwxyzABCDEFGHIJKLMNOPQRSTUVWXYZ"],
↪  "kwargs": {}}
7:   {"args":
↪  ["\ud83d\ude00\ud83d\ude03\ud83d\ude04\ud83d\ude01\ud83d\ude06->\ud」
↪  83d\ude05\ud83d\ude02\ud83e\udd23\ud83d\ude0a\ud83d\ude07"],
↪  "kwargs": {}}
8:   {"args": ["a->b,c->d"], "kwargs": {}}
9:   {"args": ["\u3053\u3093\u306b\u3061\u306f->\u4e16\u754c,\u4f60\u59」
↪  7d->\u4e16\u754c,\uc548\ub155\ud558\uc138\uc694->\uc138\uacc4"],
↪  "kwargs": {}}
10:  {"args": ["\ud83c\udf1f\ud83c\udf20\u2728\ud83d\udcab\u2b50"],
↪  "kwargs": {}}
11:  {"args": ["ZYXWVUTSRQPONMLKJIHGFEDCBA"], "kwargs": {}}
12:  {"args": ["AaBbCcDdEeFfGgHhIiJjKkLlMmNnOoPpQqRrSsTtUuVvWwXxYyZz"],
↪  "kwargs": {}}
```

```
13:  {"args": ["aaaaaaaaaaaaaaaaaaaaaaaaaaaaaaaaaaaaaaaaaaaaaaaaaaaaaaaaa↲
↪  aaaaaaaaaaaaaaaaaaaaaaaaaaaaaaaaaaaaaaaaaaaaaa"], "kwargs":
↪  {}}
14:  {"args": ["abcdefghijklmnopqrstuvwxyzABCDEFGHIJKLMNOPQRSTUVWXYZ012↲
↪  3456789!@#$%^&*()_+"], "kwargs":
↪  {}}
15:  {"args": ["aa"], "kwargs": {}}
16:  {"args": [",,,,,->...."], "kwargs": {}}
17:  {"args": ["\u03b1\u03b2\u03b3\u03b4\u03b5\u03b6\u03b7\u03b8\u03b9\↲
↪  u03ba\u03bb\u03bc\u03bd\u03be\u03bf\u03c0\u03c1\u03c3\u03c4\u03c5\u↲
↪  03c6\u03c7\u03c8\u03c9"], "kwargs":
↪  {}}
18:  {"args": ["1234567890"], "kwargs": {}}
19:  {"args": ["a->a,b->b,c->c,d->d"], "kwargs": {}}
20:  {"args": ["A1->B2,C3->D4,E5->F6"], "kwargs": {}}
21:  {"args": ["123456789"], "kwargs": {}}
22:  {"args": ["a->b,c->d,e->f,g->h"], "kwargs": {}}
23:  {"args": ["a->a,b->b,c->c"], "kwargs": {}}
24:  {"args": ["a\nb\tc\rd->e\nf\tg\rh"], "kwargs": {}}
25:  {"args": [""], "kwargs": {}}
26:  {"args": ["->->->->->"], "kwargs": {}}
27:  {"args": ["a->b->c->d->e->f->g->h->i->j->k->l->m->n->o->p->q->r->s↲
↪  ->t->u->v->w->x->y->z"], "kwargs":
↪  {}}
28:  {"args": ["\u7532->\u4e59,\u4e19->\u4e01,\u620a->\u5df1"],
↪  "kwargs": {}}
29:  {"args": ["\u6df7\u5408\u5b57\u7b26\u4e32with\u82f1\u6587and\u6570↲
↪  \u5b57123"], "kwargs":
↪  {}}
30:  {"args": ["\u0124\u011b\u013c\u013c\u00f6"], "kwargs": {}}
31:  {"args": ["!@#$%^&*()_+"], "kwargs": {}}
32:  {"args": ["aaaaabbbbbccccc"], "kwargs": {}}
33:  {"args": [".,->.,->.,->.,->.,->."], "kwargs": {}}
34:  {"args": ["\u00c4\u00d6\u00dc\u00e4\u00f6\u00fc\u00df"], "kwargs":
↪  {}}
35:  {"args": ["a->b->c->d->e->f->g->h->i->j"], "kwargs": {}}
36:  {"args": ["A->1,B->2,C->3,D->4,E->5,F->6,G->7,H->8,I->9,J->0"],
↪  "kwargs": {}}
37:  {"args":
↪  ["\ud83c\udf1f->\ud83c\udf19,\ud83c\udf1e->\ud83c\udf0d"],
↪  "kwargs": {}}
38:  {"args":
↪  ["\u03b1\u03b2\u03b3\u03b4\u03b5->\u03b6\u03b7\u03b8\u03b9\u03ba,\u↲
↪  03bb\u03bc\u03bd\u03be\u03bf->\u03c0\u03c1\u03c3\u03c4\u03c5"],
↪  "kwargs": {}}
39:  {"args": ["AaAaAa->BbBbBb,CcCcCc->DdDdDd"], "kwargs": {}}
40:  {"args": ["\u3053\u3093\u306b\u3061\u306f->\u4e16\u754c"],
↪  "kwargs": {}}
41:  {"args": [".,->"], "kwargs": {}}
42:  {"args": ["a->b,c->d,e->f"], "kwargs": {}}
43:  {"args": ["!@#$%^&*()_+-=[]{}|;:'\",.<>?/˜`"], "kwargs": {}}
44:  {"args": ["dcba"], "kwargs": {}}
45:  {"args": ["a1->b2,c3->d4,e5->f6"], "kwargs": {}}
46:  {"args":
↪  ["abcdefghijklmnopqrstuvwxyzABCDEFGHIJKLMNOPQRSTUVWXYZ0123456789"],
↪  "kwargs": {}}
47:  {"args": ["AaAaAa->BbBbBb,CcCcCc->DdDdDd,EeEeEe->FfFfFf"],
↪  "kwargs": {}}
48:  {"args": ["aaaaaaaaaaaaaaaaaaaaa->bbbbbbbbbbbbbbbbbbbbb"], "kwargs":
↪  {}}
49:  {"args": ["ABCDEFGHIJKLMNOPQRSTUVWXYZ"], "kwargs": {}}
50:  {"args": ["a\nb\tc\rd"], "kwargs": {}}
51:  {"args":
↪  ["\u0124\u011b\u013c\u013c\u00f6->\u0174\u00f4\u0159\u013c\u010f"],
↪  "kwargs": {}}
```

```
52:  {"args": ["->"], "kwargs": {}}
53:  {"args": ["\u00c4\u00d6\u00dc\u00e4\u00f6\u00fc\u00df->\u00e0\u00e
↪   1\u00e2\u00e3\u00e4\u00e5\u00e6\u00e7\u00e8\u00e9\u00ea\u00eb\u00ec
↪   \u00ed\u00ee\u00ef"], "kwargs":
↪   {}}
54:  {"args": ["\u0000->\u0001,\u0002->\u0003"], "kwargs": {}}
55:  {"args":
↪   ["\uff21\uff22\uff23\uff24\uff25->\uff26\uff27\uff28\uff29\uff2a,\u
↪   ff2b\uff2c\uff2d\uff2e\uff2f->\uff30\uff31\uff32\uff33\uff34"],
↪   "kwargs": {}}
56:  {"args":
↪   ["\u0124\u011b\u013c\u013c\u00f6->\u0174\u00f4\u0155\u0142\u0111"],
↪   "kwargs": {}}
57:  {"args": ["a"], "kwargs": {}}
58:  {"args":
↪   ["\ud83d\ude42\ud83d\ude0a\ud83d\ude00\ud83d\ude01\ud83d\ude02\ud83
↪   e\udd23\ud83d\ude03\ud83d\ude04\ud83d\ude05\ud83d\ude06"],
↪   "kwargs": {}}
59:  {"args": ["aaaaaaaaaaaaaaaaaaaaaaaaaaaaaaaaaaaaaaaaaaaaaaaaaaaaaa
↪   aaaaaaaaaaaaaaaaaaaaaaaaa->bbbbbbbbbbbbbbbbbbbbbbbbbbbbbbbbbbbbbbbb
↪   bbbbbbbbbbbbbbbbbbbbbbbbbbbbbbbbbbbb"], "kwargs":
↪   {}}
60:  {"args": ["aaaaaaaaaaaaaaaaaaaaaaaaaaaaaaaaaaaaaaaaaaaaaaaaaaaaaa
↪   aaaaaaaaaaaaaaaaaaaaaaaaa"], "kwargs":
↪   {}}
```

First 60 generated cases for _pad_version

```
0:   {"args": [["9", "8", "7", "6", "5", "4", "3", "2", "1"], ["1",
↪   "2", "3", "4", "5", "6", "7", "8", "9"]], "kwargs": {}}
1:   {"args": [["1", "2", "3", "a", "b"], ["1", "2", "3", "4", "5"]],
↪   "kwargs": {}}
2:   {"args": [["999999999999999999999999999"], ["1", "0", "0", "0",
↪   "0", "0", "0", "0", "0", "0", "0", "0"]], "kwargs": {}}
3:   {"args": [["1", "2", "3", "!@#", "$%^", "&*()"], ["4", "5", "6",
↪   "<>?", ":{}", "[]"]], "kwargs": {}}
4:   {"args": [["1", "2", "3"], ["1", "2", "3", "4"]], "kwargs": {}}
5:   {"args": [["0.1", "0.2", "0.3", "0.4", "0.5"], ["1.1", "1.2",
↪   "1.3"]], "kwargs": {}}
6:   {"args": [["1", "2", "3", "4", "5", "a", "b", "c", "d", "e"],
↪   ["5", "4", "3", "2", "1", "e", "d", "c", "b", "a"]], "kwargs": {}}
7:   {"args": [["10", "11", "12"], ["9", "10", "11", "12", "13"]],
↪   "kwargs": {}}
8:   {"args": [["10", "20", "30", "40"], ["5", "15", "25"]], "kwargs":
↪   {}}
9:   {"args": [["1", "2", "3", "4", "5", "6", "7", "8", "9", "10"],
↪   ["10", "9", "8", "7", "6", "5", "4", "3", "2", "1"]], "kwargs": {}}
10:  {"args": [["0", "0", "1"], ["0", "0", "2"]], "kwargs": {}}
11:  {"args": [[], ["1", "2", "3", "4", "5"]], "kwargs": {}}
12:  {"args": [["1", "2", "3", "a", "b", "c"], ["1", "2", "3", "4",
↪   "5", "d", "e", "f"]], "kwargs": {}}
13:  {"args": [["1.1", "2.2", "3.3"], ["4.4", "5.5", "6.6"]], "kwargs":
↪   {}}
14:  {"args": [["1", "2", "3", "4", "5"], []], "kwargs": {}}
15:  {"args": [["0"], ["0", "0", "0", "1"]], "kwargs": {}}
16:  {"args": [["1", "2", "3", "4", "5", "6", "7", "8", "9", "10"],
↪   ["1", "2", "3"]], "kwargs": {}}
17:  {"args": [["9999999999"], ["1111111111"]], "kwargs": {}}
18:  {"args": [["\u03b1", "\u03b2", "\u03b3"], ["a", "b", "c"]],
↪   "kwargs": {}}
19:  {"args": [["1", "a", "2", "b", "3", "c"], ["10", "20", "30"]],
↪   "kwargs": {}}
20:  {"args": [["a", "b", "c", "1", "2", "3"], ["x", "y", "z", "7",
↪   "8", "9"]], "kwargs": {}}
```

```
21: {"args": [["\u4f60\u597d", "\u4e16\u754c"], ["Hello", "World",
↪ "123"]], "kwargs": {}}
22: {"args": [["1", "a", "2", "b"], ["1", "2", "3", "4"]], "kwargs":
↪ {}}
23: {"args": [[], ["1", "2", "3"]], "kwargs": {}}
24: {"args": [["1", "2", "3", "alpha", "beta"], ["1", "2", "3",
↪ "gamma", "delta"]], "kwargs": {}}
25: {"args": [["\u4f60\u597d", "\u4e16\u754c", "123", "456"],
↪ ["Hello", "World", "789", "0"]], "kwargs": {}}
26: {"args": [["0", "0", "1"], ["0", "0", "0", "1"]], "kwargs": {}}
27: {"args": [[], []], "kwargs": {}}
28: {"args": [["999999999", "888888888"], ["111111111", "222222222",
↪ "333333333"]], "kwargs": {}}
29: {"args": [["1", "a", "2", "b", "3", "c"], ["4", "d", "5", "e",
↪ "6", "f"]], "kwargs": {}}
30: {"args": [["9999999999"], ["1", "0", "0", "0", "0", "0", "0", "0",
↪ "0", "0", "0"]], "kwargs": {}}
31: {"args": [["2147483647", "a"], ["-2147483648", "b"]], "kwargs": {}}
32: {"args": [["1", "2", "3", "a", "b", "c"], ["4", "5", "6", "7",
↪ "d", "e", "f"]], "kwargs": {}}
33: {"args": [["0", "0", "1", "a", "b", "c"], ["0", "0", "0", "1",
↪ "d", "e", "f"]], "kwargs": {}}
34: {"args": [["0"], ["0", "0", "0", "1", "a", "b", "c"]], "kwargs":
↪ {}}
35: {"args": [["-1", "-2", "-3", "4", "5"], ["1", "2", "3", "-4",
↪ "-5"]], "kwargs": {}}
36: {"args": [["1e10"], ["1e-10", "2e-10", "3e-10"]], "kwargs": {}}
37: {"args": [["-1", "-2", "-3", "a", "b"], ["1", "2", "3", "4",
↪ "5"]], "kwargs": {}}
38: {"args": [["1", "0", "0", "0", "0", "0", "0", "0", "0", "0"],
↪ ["9", "9", "9", "9", "9", "9", "9", "9", "9"]], "kwargs": {}}
39: {"args": [["9", "8", "7", "6", "5", "4", "3", "2", "1"], ["9",
↪ "8", "7", "6", "5", "4", "3", "2", "1", "0"]], "kwargs": {}}
40: {"args": [["0.1", "0.01", "0.001"], ["1000", "100", "10"]],
↪ "kwargs": {}}
41: {"args": [["10", "20", "30"], ["1", "2", "3", "4", "5"]],
↪ "kwargs": {}}
42: {"args": [["3.14159265358979323846"], ["2.71828182845904523536"]],
↪ "kwargs": {}}
43: {"args": [["0"], ["0"]], "kwargs": {}}
44: {"args": [["2147483647"], ["2147483648"]], "kwargs": {}}
45: {"args": [["1", "2", "3", "4", "5", "6", "7", "8", "9", "10"],
↪ ["1"]], "kwargs": {}}
46: {"args": [["0", "00", "000"], ["0", "00", "000", "0000"]],
↪ "kwargs": {}}
47: {"args": [["1", "a", "2", "b"], ["1", "2", "3", "c"]], "kwargs":
↪ {}}
48: {"args": [["1", "0", "0", "0", "0", "0", "0", "0", "0", "0"],
↪ ["9", "9", "9", "9", "9", "9", "9", "9", "9", "9", "9"]], "kwargs":
↪ {}}
49: {"args": [["1.1.1", "2.2.2", "3.3.3"], ["4.4.4", "5.5.5",
↪ "6.6.6"]], "kwargs": {}}
50: {"args": [["1", "2", "3", "4", "5"], ["5", "4", "3", "2", "1",
↪ "0", "-1", "-2"]], "kwargs": {}}
51: {"args": [["999999999999999999999999999"], ["1", "2", "3"]],
↪ "kwargs": {}}
52: {"args": [["5", "4", "3", "2", "1"], ["1"]], "kwargs": {}}
53: {"args": [["9999", "8888", "7777", "alpha"], ["1", "2", "3", "4",
↪ "5", "beta"]], "kwargs": {}}
54: {"args": [["1", "2", "3", "4", "5", "6", "7", "8", "9", "10",
↪ "11", "12"], ["1"]], "kwargs": {}}
55: {"args": [["\u03b1", "\u03b2", "\u03b3"], ["a", "b", "c", "d",
↪ "e"]], "kwargs": {}}
56: {"args": [["1", "2", "3", "4", "5"], ["1", "2", "3", "a", "b"]],
↪ "kwargs": {}}
```

```
57:  {"args": [["a", "b", "c"], ["1", "2", "3"]], "kwargs": {}}
58:  {"args": [["100", "200", "300"], ["99", "199", "299"]], "kwargs":
↪    {}}
59:  {"args": [["1"], ["1", "0", "0", "0"]], "kwargs": {}}
60:  {"args": [["10", "0", "1"], ["9", "9", "9"]], "kwargs": {}}
```

First 60 generated cases for `get_flag_suggestions`

```
0:   {"args": ["typo", ["typo1", "typo2", "correct"]], "kwargs": {}}
1:   {"args": ["ambiguous", ["ambiguous1", "ambiguous2", "ambiguous3",
↪    "unambiguous", "ambiguous"]], "kwargs": {}}
2:   {"args": ["aaaaaaaaaa", ["aaaaaaaaa", "aaaaaaaaaa",
↪    "aaaaaaaaaaa"]], "kwargs": {}}
3:   {"args": ["\u3053\u3093", ["\u3053\u3093\u306b\u3061\u306f",
↪    "\u3053\u3093\u3070\u3093\u306f",
↪    "\u3053\u3093\u306a\u306b\u3061\u306f"]], "kwargs": {}}
4:   {"args": ["", ["option1", "option2", "option3"]], "kwargs": {}}
5:   {"args": ["\u65e5\u672c\u8a9e", ["\u65e5\u672c\u8a9e",
↪    "\u4e2d\u6587", "\ud55c\uad6d\uc5b4"]], "kwargs": {}}
6:   {"args": ["no_match", ["completely", "different", "options"]],
↪    "kwargs": {}}
7:   {"args": ["a", ["apple", "banana", "cherry"]], "kwargs": {}}
8:   {"args": ["casesensitive", ["CaseSensitive", "casesensitive",
↪    "CASESENSITIVE"]], "kwargs": {}}
9:   {"args": ["prefix", ["prefix_long_option1", "prefix_long_option2",
↪    "different_option"]], "kwargs": {}}
10:  {"args": ["typo", ["type", "types", "typescript", "typoo"]],
↪    "kwargs": {}}
11:  {"args": ["flag123", ["flag123=value", "flag124=value",
↪    "flag125=value"]], "kwargs": {}}
12:  {"args": ["short", ["s", "sh", "sho", "shor", "short", "shorts"]],
↪    "kwargs": {}}
13:  {"args": ["very_similar", ["very_similar1", "very_similar2",
↪    "very_similar3", "completely_different"]], "kwargs": {}}
14:  {"args": ["!@#$%^&*", ["!@#$%^&*", "special_chars",
↪    "normal_option"]], "kwargs": {}}
15:  {"args": ["completelydifferent", ["apple", "banana", "cherry",
↪    "date"]], "kwargs": {}}
16:  {"args": ["flag123", ["flag123=value", "flag124=value",
↪    "flag125=value", "flag123"]], "kwargs": {}}
17:  {"args": ["abc", ["abcd", "abce", "abcf"]], "kwargs": {}}
18:  {"args": ["flag", []], "kwargs": {}}
19:  {"args": ["apple", ["apple", "apples", "applesauce"]], "kwargs": {}}
20:  {"args": ["verylong", ["verylongoptionname", "anotherlongoption",
↪    "yetanotherlongoption"]], "kwargs": {}}
21:  {"args": ["verysimilar", ["verysimilar1", "verysimilar2",
↪    "verysimilar3"]], "kwargs": {}}
22:  {"args": ["aaa", ["aaaa", "aaaaa", "aaaaaa", "bbb"]], "kwargs": {}}
23:  {"args": ["exact", ["exact", "exactly", "exacting"]], "kwargs": {}}
24:  {"args": ["special!@#", ["special!@#", "special$%^",
↪    "special&*()"]], "kwargs": {}}
25:  {"args": ["short", ["s", "sh", "sho", "shor", "short"]], "kwargs":
↪    {}}
26:  {"args": ["hello", []], "kwargs": {}}
27:  {"args": ["hel", ["hello", "help", "health"]], "kwargs": {}}
28:  {"args": ["longflagname", ["longflagname1", "longflagname2",
↪    "longflagname3", "shortflag"]], "kwargs": {}}
29:  {"args": ["flag=value", ["flag1=value", "flag2=value",
↪    "flag3=value", "flag=othervalue"]], "kwargs": {}}
30:  {"args": ["healh", ["health", "help", "hello"]], "kwargs": {}}
31:  {"args": ["prefix", ["prefix_option1", "prefix_option2",
↪    "different_option"]], "kwargs": {}}
32:  {"args": ["option", ["option1=value", "option2=value",
↪    "option3=value"]], "kwargs": {}}
```

```
33: {"args": ["completelydifferent", ["apple", "banana", "cherry"]],
↪    "kwargs": {}}
34: {"args": ["hlp", ["help", "hello", "health"]], "kwargs": {}}
35: {"args": ["num123", ["num1234", "num12345", "num123456"]],
↪    "kwargs": {}}
36: {"args": ["verylongflagname", ["verylongflagname1",
↪    "verylongflagname2", "verylongflagname3"]], "kwargs": {}}
37: {"args": ["hel", ["help", "hello", "health", "helmet"]], "kwargs":
↪    {}}
38: {"args": ["  whitespace  ", ["whitespace", " whitespace ", "
↪    whitespace  "]], "kwargs": {}}
39: {"args": ["completelydifferent", ["option1", "option2",
↪    "option3"]], "kwargs": {}}
40: {"args": ["mixed_case", ["MIXED_CASE", "mixed_case", "MixedCase"]],
↪    "kwargs": {}}
41: {"args": ["  whitespace  ", ["whitespace", " whitespace ", "
↪    whitespace  ", "no_whitespace"]], "kwargs": {}}
42: {"args": ["helpp", ["help", "hello", "health"]], "kwargs": {}}
43: {"args": ["option", []], "kwargs": {}}
44: {"args": ["flag=value", ["flag1=value", "flag2=value",
↪    "flag3=value"]], "kwargs": {}}
45: {"args": ["multi\nline", ["multi\nline", "multiline", "multi line",
↪    "multi\tline"]], "kwargs": {}}
46: {"args": ["test-flag", ["test_flag", "test-flag", "testflag"]],
↪    "kwargs": {}}
47: {"args": ["prefix", ["prefix_option1", "prefix_option2",
↪    "different_option", "prefixx"]], "kwargs": {}}
48: {"args": ["ambiguous", ["ambiguous1", "ambiguous2", "ambiguous3",
↪    "unambiguous"]], "kwargs": {}}
49: {"args": ["flag", ["flag1", "flag2", "flag3", "flag4", "flag5",
↪    "flag6", "flag7", "flag8", "flag9", "flag10"]], "kwargs": {}}
50: {"args": ["verylongflagname", ["verylongflagname1",
↪    "verylongflagname2", "verylongflagname3", "shortflag"]], "kwargs":
↪    {}}
51: {"args": ["mixed_case", ["MIXED_CASE", "mixed_case", "MixedCase",
↪    "mixedcase"]], "kwargs": {}}
52: {"args": ["\u3053\u3093\u306b\u3061\u306f",
↪    ["\u3053\u3093\u306b\u3061\u306f",
↪    "\u3055\u3088\u3046\u306a\u3089", "\u304a\u306f\u3088\u3046"]],
↪    "kwargs": {}}
53: {"args": ["multi\nline", ["multi\nline", "multiline", "multi
↪    line"]], "kwargs": {}}
54: {"args": ["aaa", ["aaaa", "aaaaa", "aaaaaa"]], "kwargs": {}}
55: {"args": ["abc", []], "kwargs": {}}
56: {"args": ["verysimilar", ["verysimilar1", "verysimilar2",
↪    "completelydifferent"]], "kwargs": {}}
57: {"args": ["verylongflagnamewithmorethanfiftycharacterstotest",
↪    ["verylongflagnamewithmorethanfiftycharacterstotest1",
↪    "verylongflagnamewithmorethanfiftycharacterstotest2",
↪    "shortflag"]], "kwargs": {}}
58: {"args": ["he", ["hello", "help", "health"]], "kwargs": {}}
59: {"args": ["aaaaaaaaaaaaaaaaaaaaaaaaaaaaaaaaaaaaaaaaaaaaaaaaaaaaaaaaaaa⅃
↪    aaaaaaaaaaaaaaaaaaaaaaaaaaaaaaaaaaaaaaaaaaa",
↪    ["aaaaaaaaaaaaaaaaaaaaaaaaaaaaaaaaaaaaaaaaaaaaaaaaaaaaaaaaaaaaaaaa⅃
↪    aaaaaaaaaaaaaaaaaaaaaaaaaaaaaaaaaaaaaaaaa",
↪    "aaaaaaaaaaaaaaaaaaaaaaaaaaaaaaaaaaaaaaaaaaaaaaaaaaaaaaaaaaaaaaaaaa⅃
↪    aaaaaaaaaaaaaaaaaaaaaaaaaaaaaaaaaaaaaaa",
↪    "aaaaaaaaaaaaaaaaaaaaaaaaaaaaaaaaaaaaaaaaaaaaaaaaaaaaaaaaaaaaaaaaaaa⅃
↪    aaaaaaaaaaaaaaaaaaaaaaaaaaaaaaaaaaaaaaa"]], "kwargs":
↪    {}}
60: {"args": ["typo", ["type", "types", "typescript"]], "kwargs": {}}
```
