# OpenReview forum: "EXecution-Eval: Can language models execute real-world code?"
_ICLR.cc/2025/Conference — Submitted to ICLR 2025_

### Official Review · Reviewer_o62o · 2024-10-19

**Soundness:** 2
**Presentation:** 2
**Contribution:** 2
**Rating:** 5
**Confidence:** 4

**Summary:**

This paper introduces a new benchmark EXE, focusing on testing the capability of LLMs to simulate code execution. EXE is made up of over 30000 tasks derived from 1,000 popular Python repositories on GitHub. In this scenario, LLMs need to execute code, involving operations like mathematical reasoning, logical inference, loop execution, and maintaining internal variable states. This paper provides a shallow breakdown on this. The pipeline to create EXE involves selecting and preprocessing GitHub repositories, synthesizing inputs based on function signatures, and then creating test cases (unit tests, and potentially, chaining functions tests) with the inputs. The authors claim their pipeline is automatic and capable of continuous new task generation with newest repositories to avoid test set contamination.


————after rebuttal————

The substantial revisions made during the rebuttal period addressed some of my concerns regarding model evaluation and test set contamination. The clarifications on dependency solving are also convincing. However, as the rest of my concerns are not fully addressed and the substantial revisions are making it a new paper with too many raw details without careful organizations. I think it would be better to be revised and submitted to follow-up venues like ICML. Besides, knowing the capabilities of LLMs in code executions should be the outcome of this paper, and I think current easy subset(see weakness 8) evaluation somehow weakens it.

Nonetheless, I have raised my score to 5 and presentation to 2 to praise for the authors’ efforts.

Good luck,

Reviewer o62o

**Strengths:**

1.Provide a benchmark of real-world Python code for testing LLM execution, the test cases are significantly harder and more representative for real-world usage, therefore providing a more realistic assessment of model capabilities,

2.Establish an automatic pipeline to create a real-world dataset for LLM-based code execution tasks.

3.Cover a wide range of programming concepts and can be potentially scaled up or updated with new tasks.

4.The unit-test based evaluation is correct, the authors also mention the potential to create more complicated test cases like using chaining functions.

**Weaknesses:**

## Major weaknesses:
1.Only GPT-4o and GPT-4o-mini are evaluated, contrary to the claim of evaluating **"several state-of-the-art LLMs."** Additional evaluation with different LLMs are recommended, like Claude, Gemini, Deepseek, Phi, Qwen, etc.

2.The claim of **"avoiding training on the test set"** relies heavily on the quality and effectiveness of the pipeline's ability to generate new test cases, which is not thoroughly demonstrated in the paper, no supplementary materials provided either. The Lack of supportive materials (either the benchmark itself or its creating code) to support claims about the framework's capabilities, weakens the contribution of a dataset paper.

3.The handling of import dependencies and the process of inlining required elements are not clearly explained. It's technically important here. Need clarification.

4.A bit limited to Python code, which may not represent the full spectrum of programming challenges across different languages. Since LLMs are pretrained on various programming languages, it's worth to know the execution capability on other programming languages.

5.Poor quality of figures in the paper, with low-precision images that are difficult to see clearly, the authors should use vector figures instead of jpgs or pngs,

6.The appendix uses 8 pages to show an example, which is excessive and poorly organized, besides, it's still not intuitive for understanding. This needs significant revision for clarity and conciseness.

## Minor weaknesses:

7.A bit limited evaluation metrics, using only Pass@1 accuracy. Considering more evaluations on Pass@k, or try some self-correction mechanism with LLM.

8.Filtering on limited acceptable types and functions seems to make EXE an **easy subset of the real real-world programs**, although it is a fair design choice for a benchmark to avoid environment configuration issues. I think it's more interesting to know the capabilities and limitations of LLMs when executing harder cases, containing real-world types like numpy.array, torch.tensor for example. Can the authors add some discussions about their findings here?

## Typos and Presentation Issues:

Line 294: tense issues, ...**increase** task difficulty, however bit manipulation and boolean operations only **showed**...  Should use unified tense throughout a paragraph.

Line 297: however for loops **on (73 Pass@1) on** average did not have a significant impact.

Line 303: Incorrect spacing on the title of the rightmost subfigure.

Figure 7: Examining only on LLM really executed code makes the accuracy normal now. However, it seems the results are not clearly illsutrated (only a small part of the figure is valid now, which is not clear). Consider to use some new figures.

Appendix A.2: These are important part of your paper, since current version only uses 8 pages, consider to move this section to the main page and explain them with more details.

**Questions:**

1.In the appendix, could you clearly differentiate between original code, imported dependencies, and LLM execution steps? Can you show the full LLM output and indicate at which steps they fail?

2.How does EXE compare to existing code execution benchmarks in terms of task diversity and difficulty when "executing code"?

3.Can you elaborate on the measures taken to ensure the generated test cases are meaningful, diverse, and correctly assess code execution abilities?

4.How do you validate that the newly generated test cases are indeed novel and not present in existing LLM training sets?

5.Has the chaining-function been implemented now? Because i think it will be of more interests to the community if EXE can create more complicated test cases automatically.

---

> ### Comment · Reviewer_o62o · 2024-12-03
>
> Similar to Reviewer VhJ6, I don't hear the responses from authors either.  It would be good to have discussions on paper itself or future revisions. Please notice that the discussion period is coming to an end.

---

> ### Author Response · Authors · 2024-12-03
>
> Thank you for your thorough and constructive feedback. We appreciate the detailed analysis and have addressed your concerns with substantial revisions:
>
> 1. Model Evaluation
>
> We've expanded our evaluation to include both open-source models (Llama 3.1 8B/405B, Mistral-Large) and additional closed-source models (Claude 3.5 Sonnet).
>
> 2. Test Case Generation
>
> Acknowledging this we've added:
> - Empirical evidence of novel test case generation (see Fig. 6), demonstrating continuous generation of >300 unique validated cases per function
> - Appendix A.2 - Test case generation methodology
> - Appendix A.4 - Validation pipeline with seven specific validators
> - Appendix A.8 - Additional experimentation on diversity of generated test cases
>
> 3. Dependencies handeling
>
> We've added Appendix A.3 providing a detailed walkthrough of our dependency resolution and AST analysis process (inc. function selection, symbol analysis, import handling, dependency graph construction and inlining)
>
> 4. Programming Languages
>
> While we agree that multi-language evaluation would be valuable and provide additional scalability, we have focused initially on python from a pragmatic lens, and from a prior that multi-lingual isn't necessary to realise the goal of measuring long multi-step reasoning.
>
> However, we do believe the underlying methodology could extend to other languages (potentially utilising tree-sitter instead of python's ast), and hold an interest in exploring this.
>
> 6. Appendix Formatting
>
> We agree and have added a simplified example (for quick grokking) prior to the full example.
>
> 8. Filtering Third-Party Packages
>
> During EXE's development we experimented with a secondary branch - one that had a manually curated subset of the 1,000 most common 3rd-party identifiers allowed (i.e. numpy, jinja2, pydantic - unfortunately torch did not make the cut). We did not observe major accuracy shifts and hence prioritised stdlib only, however, we agree with the assessment and intend to revisit this.
>
> ### Unaddressed Issues
>
> We sincerely apologise for the presentation issues, while we've increased image resolution, we acknowledge that vector formats would be preferable.
>
> Furthermore, due to time constraints we haven't implemented Pass@k metrics, though we agree they would provide additional value. Similarly, not all tense inconsistencies and spacing issues have been addressed.
>
> ### Responses to Specific Questions
>
> 1. Code Execution Details
>
> We've restructured Appendix A.3 to clearly differentiate between original code, dependencies, and inlined code for execution. We would have further shown failure modes in detail (by vast majority, failures are predictions of the right type, with the wrong value) but have prioritised running the broader spectrum of models during time constraints.
>
> 2. Benchmark Comparison
>
> As mentioned in prior discussion regarding CruxEval and CodeNet - EXE isn't based on synthetic or competition code, hence it holds significantly more diversity in problem length, problem space, syntax used, and net difficulty.
>
> 3. Test Case Quality
>
> We've added A.2 and A.4, on test case generation. Ensuring test cases pass:
> - Schema Conformance: Validates test cases parse as valid function inputs
> - Duplication: Ensures each test case input is unique
> - Coverage: Requires minimum 10 test cases per function
> - Non-triviality: Less than 50% of cases can return unmodified input
> - Output Diversity: No single output in >66% of cases
> - Error Balance: Exception cases limited to 50% of total
> - Runtime Execution Bounds: CPU time under 10 seconds per case
>
> Furthermore in the discussion section on "ExecEval provides a..." we have demonstrating diversity through runtime metrics (see A.5) including "lines executed" and "branches taken" for test cases within the same function group.
>
> 4. Novel Test Cases
>
> While proving non-contamination of LLM-generated content is challenging, our manual searches have found: a) minimal evidence of existing test cases for functions in EXE - while some functions had unit tests, they were rare, and b) minimal evidence of our specific input-output pairs - while we could find some matching inputs, such as for simple string functions, these appeared as standalone strings (not paired with their outputs in a way enabling contamination). As further evidence of our ability to generate novel cases, we demonstrate in Fig.6 production of unique, validated test cases well beyond 300 instances per function. Suggesting the generative space for valid test inputs is sufficiently large to support near term expansion of EXE.
>
> 5. Function Chaining
>
> We've performed exciting initial experiments with function chaining and observed promising results. However, due to time constraints, we haven't completed the full evaluation we would want to include in the paper. We believe this represents a valuable future direction for expanding EXE's capabilities.
>
> We appreciate your feedback in helping us strengthen this work and are happy to provide any clarifications.

---

### Official Review · Reviewer_VhJ6 · 2024-10-29

**Soundness:** 2
**Presentation:** 2
**Contribution:** 2
**Rating:** 3
**Confidence:** 3

**Summary:**

The authors introduce a dataset of executable python functions mined from Github. The functions chosen have certain type annotations for which test cases can be generated. The task consists in providing a code snippet as well as the input arguments into an LLM and asking the LLM to predict the output (this task has been referred to as "program induction" in some literature of the past, and I will refer to it as "program understanding")

The authors argue that this is a non-trivial benchmark and that the methodology allows the benchmark to evolve over time to include test cases or functions that are not in the training set. The authors also argue that this program understanding task could be an useful gauge of LLMs performance for coding tasks.

The authors evaluate GPT4o and GPT4o-mini on this task and provide some analysis on performance by certain proxies for ``difficulty" such as lines of code, number of function calls, etc.

**Strengths:**

The authors deserve credit for their creative use of open-source software on Github. I believe that more executable coding benchmarks will be beneficial to the community and the authors have elements to create something very interesting! The steps taken to create the dataset seem non-trivial and the scale of the dataset is notable (>30K functions). There is preliminary evidence that the task is non-trivial, and the authors also have interesting analysis on factors that lead to more difficult program understanding on this task. I think there is potential for the authors to leverage their ingenuity in constructing this dataset for interesting applications. After skimming CruxEval which seems to propose a similar approach, my judgment is that the underlying dataset scale and difficulty of EXE is more noteworthy.

**Weaknesses:**

I think the motivation of this paper is great, and the creativity to create an executable programming benchmark is excellent! I think there is great potential in this work! I would recommend the authors try to focus on some of the following facets.

1. Clarification of Test case generation methodology

I may have missed it, but I tried to look for details on methodology of test case input generation. The authors are clear on the accepted types are allowed for inputs/outpus, but it is unclear how generation is done. The best I could find is: "Based on the type definition (used for setting the function calling schema) inputs/ output pairs have been generated with the goal of maximising diversity of control flow paths within the function." and "Using the argument type annotations we construct a LLM function calling schema that generates a diverse set of inputs." The paper requires more details and clarification on this, and depending on the methodology chosen, this could affect the merits of the approach.

2. Experiments / Lack of Models Considered

Because this is a datasets and benchmarks paper and the paper's motivation emphasizes "difficulty" of the task, not enough is done to substantiate this claim. My expectation for a dataset/benchmark paper should be at least to evaluate numerous open source models (e.g. CodeLLama, LLama3 family, CodeT5, etc) of varying sizes in addition to commercial models. Additionally, only 2 commercial models from OpenAI are used. Performing wider evaluation will strengthen these claims and the analysis, otherwise, it is an open question on how other models would perform on this task.

3. The framing of experiments + context of other works (a potential lack of novelty)

The authors do not distinguish their approach or experiments from a dataset like CodeNet. The code understanding experiments provided here can also be done with CodeNet. If the authors could show that LLM performance or the nature of LLM performance is different on their task vs. CodeNet, this would substantiate the contribution. Of course the code on github is more diverse in nature, but on the other hand, the input/output types are still limited, and a dataset like CodeNet is multi-lingual.

My recommendation would be to consider other creative uses of this dataset besides the ones you currently have.


4. Polished Writing

A paper for this venue should have a higher standard of polishing. For example, the term AST should be introduced as an Abstract Syntax Tree (AST) and referred to as AST. At one point the authors colloquially refer to evaluation benchmarks as "evals." These are minor points and easy to fix, but are nevertheless are standards.

5. Clarification on Licensing, Copyright, etc.

I did not see clarification if the authors filtered code for permissively licensed software and if the dataset falls under acceptable use of the software.

**Questions:**

1. Can you provide more detail on the methodology of test case input generation? Even the code used for this will work, although an explanation would help as well.

2. Can you provide more explicit clarification on your proposed contributions, especially in context of a dataset like CodeNet. Besides the fact that the functions are from Github and that the dataset in theory can evolve, is there anything else I have misunderstood?

3. [Recommendation to address limitation] Are you able to provide more comprehensive evaluations of other model? If the authors have access to computing resources, I strongly recommend open access models like CodeLlama to avoid API costs. If the authors have access to API credits, I would recommend at least one very large commercial model such as sonnet 3.5 or Llama3.1 405 Instruct (e.g. hosted on AWS bedrock). Although alone, I do not think these will convince me the paper should be accepted.

4. Licensing / Copyright: Can you explain what licenses exist for the data mined for the benchmark? e.g. was filtering done for permissive licenses? Additionally if more context can be provided then if the dataset is a fair and acceptable use of the software under consideration.

5.Clarification on Side-Effects, Determinism, and Execution Environment: Can you explain how you implement ensuring that there are "no side-effects" and that determinism indeed holds? I understand there are some banned imports, but can you provide more clarification? How do we know that this is indeed comprehensive enough to make these claims? Additionally, can you specify the python version / environment used for executing the python code? In a perfect world, it would be good to have a docker container with the same environment used to execute these programs so that the input/output examples are indeed reproducible.

---

> ### Author Response · Authors · 2024-12-03
>
> Thank you for you detailed review. We appreciate your careful reading and constructive feedback. We have made substantial revisions to address your concerns:
>
> 1. Test Case Generation:
>
> We have significantly expanded our documentation, including two specific appendices on test case generation:
> A.2 - Test case generation - schema construction, generation, and validation
> A.4 - Test case validators - schema conformance, duplication checking, coverage requirements, non-triviality checks, output diversity, error balance, and runtime bounds
>
> 2. Model Evaluation:
>
> We've expanded our evaluation to include both open source (Llama 3.1 8B/405B, Mistral-Large) and closed source models (Claude 3.5 Sonnet).
>
> 3. Differentiation from CodeNet:
>
> We appreciate the need to clarify our distinct contributions. While CodeNet provides a valuable dataset, EXE differs fundamentally:
> - While CodeNet contains a massive 13,916,868 unique pieces of code, these map to only 4,053 unique problems with the same test case set, sourced only two competitive programming websites.
> - This creates a significant contamination risk and poses a question of how it could scale beyond competitive coding problems to a more general set.
> - To demonstrate this concretely, we tested the memorisation of random CodeNet problems. For example, asking Claude "What is problem ALDS1_5_A on AIZU?" [1] produces a perfect description and implementation purely from memory:
> ```
> From what I understand, ALDS1_5_A on AIZU ONLINE JUDGE is the "Exhaustive Search" problem, which is also sometimes called the "Sum of Subset" problem. Here's what I believe the problem entails:
>
> Given:
> - A sequence A of n integers
> - A sequence of q queries, each query being an integer m
> - 1 ≤ n ≤ 20
> - 1 ≤ q ≤ 200
> - All input integers are positive and less than 2000
>
> The task is to determine for each query m whether it's possible to select some elements from sequence A such that their sum equals m.
>
> I can write a solution in Python that should solve this problem efficiently:
>
>
> def can_make_sum(A, target):
> n = len(A)
> # Try all possible combinations using bit manipulation
> for i in range(1 << n):
> sum = 0
> for j in range(n):
> if (i & (1 << j)):
> sum += A[j]
> if sum == target:
> return True
> return False
>
> # Read input
> n = int(input())
> A = list(map(int, input().split()))
> q = int(input())
> queries = list(map(int, input().split()))
>
> # Process each query
> for m in queries:
> print("yes" if can_make_sum(A, m) else "no")
>
>
> This solution uses bit manipulation to try all possible combinations of elements from the sequence A. For each query m, it checks if any combination sums to m.
>
> Since n ≤ 20, trying all 2^n combinations is feasible. The solution has a time complexity of O(2^n)...
> ```
> - In contrast, EXE sources from a broad range of evolving GitHub repositories. Generating distinct tasks, with unique execution paths, on a per-needs basis.
> - Furthermore EXE has non-static test cases, Fig. 6 (demonstrating generation of >300 novel test cases per function), suggests our approach can significantly expand (20x min) where CodeNet is limited by competitive programming problems
> - Finally, our objective differs - EXE's is using code execution as a proxy for measuring long multi-step reasoning, to do so requires EXE's tailoring to (a) have a broad representation of problems from the real world, (b) generate enough diverse test cases to test reasoning on those problems.
>
> 4. Writing Polish: We have improved technical precision throughout
>
> 5. Licensing:
>
> During the repo collation process we filtered to repositories including only those with permissive licenses allowing derivative works with or w/o attribution. To confirm our filtering we performed a re-review of all packages and test cases. During this, 3 seperate packages (with 75 test cases) were identified as not having permissive licences, these have been removed, causing the dataset size difference between the two paper revisions.
>
> 6. Technical Implementation:
>
> We've added significantly more technical documentation in the appendices, specifically:
> - A.3 - Details function selection, dependency resolution and AST analysis, with A.3.3 discussing filtering of functions post inlining
> - However, with greater time constraints we would like to further discuss the methodology including:
>   - Our lengthy manual review of all python stdlibs for side effect containing modules, classes and functions,
>   - Our methodology for selecting the most permissive removals (i.e. one function having a side effect should not stop usage of the module, only that function),
>   - Our lengthy manual review of all tasks
> - Python 3.12.2 is used
>
> The revised paper maintains the core contributions while providing significantly more technical depth and experimental validation. We believe these changes address your concerns while strengthening the paper's contributions. We're happy to provide any additional clarification needed.
>
> [1]: https://judge.u-aizu.ac.jp/onlinejudge/description.jsp?id=ALDS1_5_A

---

### Official Review · Reviewer_BGbV · 2024-11-01

**Soundness:** 3
**Presentation:** 3
**Contribution:** 2
**Rating:** 5
**Confidence:** 3

**Summary:**

The authors present a new benchmark to evaluate LLMs' capability in executing real-world code. To collect a set of executable code from the real world, they built a pipeline to collect repos from GitHub to construct self-contained, deterministic code. They performed static analysis to inline the dependencies to make it self-contained, and then generated inputs using LLMs. The benchmark includes 30,000 tasks across 1,000 popular Python repos. They evaluated GPT-4o and GPT-4o mini and showed that these strong models still struggle with more complex tasks.

**Strengths:**

* The benchmark addresses the issue in the prior work, i.e. CruxEval, by collecting real-world Python functions, instead of synthetically generated ones from LLMs.
* The benchmark includes diverse tasks and spans across 1000 repos
* The pipeline is mostly automatic and can be updated to include newer repos to address the benchmark contamination problem
* They provide analysis regarding the relationship between performance and line count, number of function calls, execution time, etc. to better understand what affects performance

**Weaknesses:**

* The main issue with the work is that it lacks certain insights as to how this benchmark would shed light. For example, many people use CruxEval because it correlates well with model's code generation/understanding ability. Does evaluating on this benchmark instead of CruxEval serve as a better predictor of such capability?
* The paper evaluates on two models: GPT4o and GPT4o-mini. It would be better to also evaluate some open source models to compare against the closed API-only ones, especially the StarCoder model which explicitly provides training data, so one can check whether the code in the training data affects the execution prediction or not
* The input test cases are LLM generated. Since the work emphasizes real-world scenarios, it would be good to assess whether the LLM-generated test cases are of reasonable quality, and whether it gives an advantage to the LLM that generated the test cases in performing the task

**Questions:**

* What is the model used to generate inputs? Does it matter if different models are used for input generation?
* The inlining to create a doable Python program, although necessary to make the task self-contained, also seems to make the code not look like real-world cases. Is there a way to address this?
* Are there any observations on what types of packages the LLM struggles with? Is there more we can learn if there is more thorough error analysis?

---

> ### Author Response · Authors · 2024-11-24
>
> Thank you for you detailed review. We appreciate your careful reading and constructive feedback. We have made substantial revisions to address your concerns:
>
> 1. Regarding benchmark insights and predictive ability:
>
> While models like GPT-4o achieve high performance on CruxEval (89.2%), we observe interesting discrepancies between CruxEval scores and real-world performance. For instance, Claude-3.5 Sonnet often demonstrates stronger capabilities in benchmarks like Aider despite lower CruxEval scores, suggesting potential limitations in predictiveness [1][2]. However, our primary contribution does not target creating a more predictive benchmark. Instead, we target demonstrating:
>
> 1.1. That artificially limited function sets and LLM generated code problems are no longer necessary - we can construct and evaluate on complex, messy, real-world code.
>
> 1.2. That there is potential to significantly scale difficulty with this "style" of evaluation - we show strong correlations between task difficulty and largely controllable factors e.g. execution steps and identifier counts.
>
> 1.3. That this dataset can be scaled cost-effectively - EXE's dataset generation and evaluation pipeline costs under $150 to run.
>
> 2. On models:
> We've expanded our evaluation to include both open source (Llama 3.1 8B/405B, Mistral-Large) and closed source models (Claude 3.5 Sonnet).
>
> 3. On test case quality:
>
> We have added A.2 and A.4, describing test case generation and quality control. In brief, test cases must pass:
> - Schema Conformance: Validates test cases parse as valid function inputs
> - Duplication: Ensures each test case input is unique
> - Coverage: Requires minimum 10 test cases per function
> - Non-triviality: Less than 50% of cases can return unmodified input
> - Output Diversity: No single output in >66% of cases
> - Error Balance: Exception cases limited to 50% of total
> - Runtime Execution Bounds: CPU time under 10 seconds per case
>
> We are interested in performing comparison experiments between models for input generation & output generation, this represents an interesting future work direction.
>
> Questions:
>
> Q1:
>
> We use GPT-4o for generation, spanning both 2024-08-06 and 2024-05-13 versions. We have only performed light testing of test case generation with differing models thus far. During which we noted significant differences in prompting required to generate strong edge cases - Claude 3.5 sonnet for example only required an engineered prompt tens of tokens long to match gpt4o with several hundred tokens. We're interested in further exploring these differences.
>
> Q2:
>
> As stated in our added Appendix A.2. on input generation, we are motivated to inline as it ensures the language model executes exactly the same code as our interpreter. However we recognise the trade-off between giving a model the exact code to be run and giving it the code how a human is likely to read it. While we've selected the side that we initially believed would be the most fair (in asking a model to perform a faithful execution), we've now begun exploring alternatives, including a multi-file output where source files are merged with explicit origin tags. We believe comparing these approaches could prove valuable.
>
> Q3:
>
> We performed manual inspection on many of the lowest scoring packages and have provided some of our notes:
> - nltk: unexpected failures - EXE extracted stemming operations and n-gram matching - surprisingly even high-performing models like Claude and GPT-4o performed poorly at stemming operations, potentially due to tokenization
> - cryptography: EXE extracted modular inverses, greatest common denominators, hex conversions, to give a few examples.
> - mdurl: EXE extracted many complex bit manipulation functions for UTF-8 processing, to illustrate the challenge:
>   ```python
>   if (b1 & 0xF8) == 0xF0 and (i + 9 < l):
>       b2 = int(seq[i + 4 : i + 6], 16)
>       b3 = int(seq[i + 7 : i + 9], 16)
>       b4 = int(seq[i + 10 : i + 12], 16)
>       if (b2 & 0xC0) == 0x80 and (b3 & 0xC0) == 0x80 and (b4 & 0xC0) == 0x80:
>   ```
> - wasabi: EXE extracted precise text wrapping and diff generation functions, requiring exact character-level manipulation
> - statsmodels: EXE extracted mathematical operations with hardcoded statistical constants
>
> We believe there is great potential for further qualitative analysis and cross package identification of common failure modes. As mentioned in Section 3, many well known issues have been observed; floating point operations, new syntax, bit operations. However, we have also observed new patterns, for example, read vs write with variables - the magnitude of newly defined variables has a far smaller effect on pass rate than the magnitude of how many times those variables were used elsewhere.
>
> We appreciate your feedback in helping us strengthen this work and are happy to provide any clarifications.
>
> [1]: Qwen2.5-Coder Technical Report - https://arxiv.org/pdf/2409.12186
> [2]: Aider LLM Leaderboards - https://aider.chat/docs/leaderboards/

---

> > ### Comment · Reviewer_BGbV · 2024-11-26
> >
> > Thank you for your detailed response.
> > It is great that the work includes very comprehensive evaluations of many open and API-only models. I think this is valuable data, especially on complicated code execution tasks.
> >
> > I think the main issue with the work is still the lack of clear motivation or insights - why is evaluating complicated code execution for LLMs useful? I certainly think it could be very useful; for example, CruxEval is used widely to track LLMs' code understanding capability, and it often correlates with the models' code generation capability. I think including more of this kind of useful analysis in future work could make this work better.

---

> > > ### Author Response · Authors · 2024-12-04
> > >
> > > Thank you, we appreciate your feedback and are glad that we've been able to address some concerns.
> > >
> > > Regarding motivation and insights:
> > > - In the spirit of avoiding readers having to read multiple similar replies, we have replied in depth to EDiE regarding motivation and would request readers review that comment first.
> > > - In addition to our prior reply in this same thread, we have since become aware of how rapidly CruxEval is approaching saturation.
> > >     - OpenAI's o1-mini [1] has scored 96.2%, leaving little further room for comparison of models and demonstrating how quickly restrictive, synthetic benchmarks can approach saturation.
> > >     - We believe this reenforces the need for broader, real-world datasets - to enable the opportunities highlighted in our motivations, and maintain the valuable insights datasets such as CruxEval have previously provided.
> > >
> > >
> > > We greatly appreciate the reviewers' emphasis on clarifying our motivations. Your feedback has helped us articulate these points more effectively. If you have remaining concerns about any aspect of our motivation or approach, we welcome specific questions in your final notes to ensure we've fully addressed all points of uncertainty.
> > >
> > > [1]: Qwen2.5-Coder Technical Report - https://arxiv.org/pdf/2409.12186

---

### Official Review · Reviewer_EiDE · 2024-11-04

**Soundness:** 1
**Presentation:** 3
**Contribution:** 2
**Rating:** 3
**Confidence:** 4

**Summary:**

The paper introduces EXE, a new benchmark designed to evaluate language models (LLMs) on their ability to execute Python code sourced from real-world applications. This benchmark aims to address several limitations of existing evaluations, particularly the issues of scalability, task diversity, training data contamination, and benchmarking costs.

The benchmark comprises over 30,000 tasks drawn from 1,000 popular GitHub repositories, spanning different complexities and computational operations like logical inference, mathematical reasoning, and state management.

To construct this benchmark, the authors first select the top 1,000 most popular pypi packages and collate the corresponding github repos, after that, the authores perform a static ast analysis to filter to functions with LLM generatable argument and return type annotations. Finally, the authors apply LLM to generate test cases.

The evaluation with GPT-4 model demonstrate the limitation of existing code models.

**Strengths:**

### 1. This paper is well-written and easy to follow.

### 2. Benchmarking code LLM is an important problem.

### 3.  The findings are interesting.

**Weaknesses:**

## 1. The motivation for this work is not clearly articulated.

The paper proposes benchmarking the code execution capabilities of LLMs, but it is unclear why such a capability is needed given the existing roles of compilers and interpreters. A possible motivation might be that LLMs are more lightweight and could predict execution outcomes without running the code. However, I did not see any evaluation results to support this assumption.

## 2. The paper suggests that the proposed dataset can guard against data contamination [1, 2], but lacks a detailed explanation of how this is achieved.

The authors claim that the dataset is dynamically collected from GitHub, which could help mitigate contamination. However, since the benchmark is built from popular GitHub repositories that do not frequently change, the dataset may not be as dynamic as implied. Additionally, because the test inputs are generated by LLMs, it is unclear how this setup effectively prevents data contamination.

## 3. Certain methodological details are missing.

First, in "Function Selection and Dependency Collation," the authors mention using static AST analysis, but it is not clear how this process is performed. Second, regarding the error metric, the authors state that they "compare the type and message (excluding stacktrace) using a language model comparison," which is described too vaguely to understand how this metric is actually computed.

## 4. This work lacks soundness in the following areas:

(1) The authors claim the benchmark is diverse; however, there is no diversity evaluation regarding the prompts and solutions. (2) Since all test cases are generated by an LLM, there is no guarantee that the test cases are sound or appropriate for the programs. Given that some test cases result in errors during execution, this raises soundness concerns.

## 5. Minor: Some figures are of low resolution and unclear.


[1] GSM-Symbolic: Understanding the Limitations of Mathematical Reasoning in Large Language Models

[2] PPM: Automated Generation of Diverse Programming Problems for Benchmarking Code Generation Models

**Questions:**

1. Why we need to benchmark LLM's executation capability.

2. Can you introduce more details of the approach and the evaluation?

---

> ### Author Response · Authors · 2024-11-24
>
> Thank you for you detailed review. We appreciate your careful reading and constructive feedback. We have made substantial revisions to address your concerns:
>
> 1. On motivation:
>
> Although the concept of replacing real parts of a code execution pipeline is rather interesting, our primary motivation differs.
>
> We see an exciting opportunity to leverage code execution as a method for studying language models' multi-step reasoning capabilities. Code execution provides a unique environment where we can reliably generate massively long and complex multi-step problems. Problems that are deterministic, measurable at each step, and can be generated at a cost basis in cents. It also offers a mechanism to generate continually new tasks, aiding with contamination.
>
> 2. On data contamination:
>
> We agree that this deserves greater clarity in our presentation. Our approach has two components: GitHub's natural evolution provides a secondary mitigation, but our primary defence lies in the continuous generation of novel test cases.
>
> While proving non-contamination of LLM-generated content is challenging, our manual searches have found: a) minimal evidence of existing test cases for functions in EXE - while some functions had unit tests, they were rare, and b) minimal evidence of our specific input-output pairs - while we could find some matching inputs, such as for simple single-string functions, these appeared as standalone strings rather than paired with their outputs in a way that would enable contamination. As further empirical evidence of our ability to generate novel cases, we demonstrate in Figure 6 continuing production of unique, validated test cases well beyond 300 instances per function. Suggesting the generative space for valid test inputs is sufficiently large to support near term expansion of EXE.
>
> 3. On methodological details:
>
> We have added documentation in Appendices:
>
> - A.3 - Details function selection, dependency resolution and AST analysis
> - A.7 - Details error comparison
> - A.2 - Details test case generation - schema construction, generation, and validation
> - A.4 - Details test case validators
> - A.6 - Details static and runtime statistics
> - A.8 - Details additional experimentation on diversity of generated test cases
>
> 4. On diversity and soundness:
>
> 4.1 Task Diversity:
> Addressing diversity in prompts and solutions we analyse both at the function and test case level:
>
> | | Function Level | Test Case Level |
> |---|---|---|
> | **Prompt (Input) Diversity** | **Q:** Are our set of functions diverse for a model to reason through? **Evidence:** Numerics & discussion provided in Section 2.2. (e.g. in "Long multi-step problems..." and "Diversity of Inputs...") | **Q:** Are our test cases diverse in requiring differing code paths and varying amounts of compute? **Evidence:** Numerics & discussion provided in Section 3 (e.g. in "ExecEval provides a..." and "Test case generation...") |
> | **End-to-end Solution (Outcome) Diversity** | **Q:** Are model outputs (pass rate) diverse across the space of functions? **Evidence:** Visuals & discussion surrounding Fig. 4,5,6, and 7 | **Q:** Are model outputs (pass rate) diverse across test cases for a single function?  **Evidence:** Numerics & discussion provided in Section 3 (e.g. in "ExecEval provides a...", "Test case generation...") |
>
> 4.2 Soundness:
>
> 4.2.1 Note on errors:
>
> Error prediction represents one of the more complex control flow structures in Python. Hence, we explicitly prompt the LLM in our generation pipeline for error-triggering inputs as they provide coverage of real world scenarios, often more challenging than standard execution flows and relevant to the cybersecurity domain.
>
> 4.2.2 Test case soundness:
>
> We have added A.2 and A.4, describing test case generation and quality control. In brief, test cases must pass:
> - Schema Conformance: Validates test cases parse as valid function inputs
> - Duplication: Ensures each test case input is unique
> - Coverage: Requires minimum 10 test cases per function
> - Non-triviality: Less than 50% of cases can return unmodified input
> - Output Diversity: No single output in >66% of cases
> - Error Balance: Exception cases limited to 50% of total
> - Runtime Execution Bounds: CPU time under 10 seconds per case
>
> 5. On figure quality: We apologise and have updated figures.
>
> Questions:
>
> Q1:
> Beyond our aforementioned motivation of measuring multi-step reasoning, EXE provides unique insights into:
> 1. Models' handling of messy, real-world code
> 2. The potential to scale difficulty with this style of evaluation
> 3. Error prediction
>
> Q2:
> We've expanded documentation as mentioned in (3), and enhanced our discussion with:
> 1. Different models (Llama 3.1 405B, Llama 3.1 8B, Mistrial Large and Claude3.5 Sonnet)
> 2. Test case generation experiments (scalability & diversity)
> 3. Statistics analysis (Appendix A.6)
>
> We believe these changes address concerns while strengthening the paper. We're happy to provide any clarifications required.

---

> > ### Comment · Reviewer_EiDE · 2024-12-02
> > **Review comments**
> >
> > Thank you to the authors for their detailed explanation and evaluation results. However, the provided classifications do not fully address my concerns.
> >
> > 1. **Motivation**: While I agree that leveraging code execution as a method for studying language models' multi-step reasoning capabilities is a reasonable approach, I still do not fully understand how these benchmarks can be effectively utilized for future work. I suggest the authors further elaborate on the relationship between "benchmarking code execution capability" and improving code models. For example, can developer utilize the benchmarking results to select a more l for a specific task?
> >
> > 2. **Data Contamination**: The authors mention that their primary defense lies in the continuous generation of novel test cases. I recommend providing more details on the methods used to generate diverse test cases and the criteria or metrics used to evaluate their diversity.
> >
> > Considering the strengths and weaknesses, I will maintain my original score.

---

> > > ### Author Response · Authors · 2024-12-04
> > >
> > > Thank you for your follow-up feedback. We appreciate the opportunity to clarify these points:
> > >
> > > ### Regarding Motivation and Future Work Applications:
> > >
> > > We have faced personal challenges incentivising us to study long multi-step reasoning (and hence build EXE):
> > > 1. It is far too difficult for downstream LLM consumers (developers, data-scientists and general users) to understand if a LLM can solve their problem or not:
> > >     - When it is possible to find a representative dataset, it is often either already saturated/ contaminated, or will rapidly become so. The time investment for a consumer to understand, build confidence and get value out of a dataset requires it to have a long shelf life.
> > >     - Overall dataset accuracy and failure modes are opaque, and are not interpretable in a way that a consumer can understand (or can construct their system to mitigate).
> > >         - To illustrate, they do not show interpretable equivalents of "accuracy fall-off with increasing reasoning steps", or "accuracy fall-off with number of variables that have to be kept in mind" - these are the type of statistics that can inform a user in decisions like "how many subpoints can I have in a legal obligation before my problem is not solvable, or when I'd need to break my solution down to run per subpoint".
> > >         - When reasoning/ agentic datasets have collected this data it has often been prohibitively expensive to do so - for example SWE-Bench-Verified - it required multiple humans to perform each and every task, some taking hours per task, to generate four low resolution buckets of 15 minutes, 1 hour, 4 hours and greater.
> > >         - Furthermore it is often hard to interpret how a new model will impact their existing problem, contamination and the lack of high resolution interpretable statistics doesn't give much insight into if there is a meaningful opportunity for them.
> > >     - These are many of the challenges that have motivated EXE's design and statistics - with the goal of providing significantly more interpretable figures that consumers are able to use.
> > > 1. It is also far too difficult today for those training/ researching models to obtain long context reasoning problems to train and test on:
> > >     - There are not many multi-step reasoning datasets for training (or long context data points [1]), and the creation of new datasets can be prohibitively expensive. EXE offers an opportunity to create many long context, challenging reasoning problems -in an area where todays models perform poorly.
> > >     - EXE's bank of code execution tasks are deterministic and can be run to provide unambiguous ground truth at every step, allowing researchers to precisely identify where reasoning breaks down in a way that is extremely challenging with traditional natural language tasks. This deterministic step-by-step nature can aid in identifying precise reasoning failure points, understanding how mechanistically models handle state tracking and multi-step logic - amongst many other investigations.
> > >     - These challenges also have motivated EXE's design and outcomes - with the goal of improving long context and multi-step reasoning through mechanistic understanding and better training sets.
> > >
> > > .
> > > ### Regarding test case diversity, we employ both quantitative metrics and validation criteria:
> > >
> > > Diversity Metrics (calculated across the 15 test cases per function, then aggregated across all functions):
> > > - For conditional statements executed, the standard deviation is 61% of the mean count, implying test cases are exercising significantly different code paths even within single functions
> > > - For lines of code executed, the standard deviation is 20% of the mean, indicating meaningful variation in execution length
> > > - Model performance shows the standard deviation of pass rates is 39% of the mean, suggesting our test cases present varying levels of challenge
> > >
> > > Validation Requirements (possibly this was missed in our prior reply, we have documented our methods further in A.2 & A.4). All test cases must pass:
> > > - Schema Conformance: Validates test cases parse as valid function inputs
> > > - Duplication: Ensures each test case input is unique
> > > - Coverage: Requires minimum 10 test cases per function
> > > - Non-triviality: Less than 50% of cases can return unmodified input
> > > - Output Diversity: No single output in >66% of cases
> > > - Error Balance: Exception cases limited to 50% of total
> > > - Runtime Execution Bounds: CPU time under 10 seconds per case
> > >
> > > Furthermore we have systematically measured diversity through runtime and static metrics (A.5). Our recently added Figure 6 demonstrates the continuous generation of novel test cases (>300 per function) while maintaining the diversity validation requirements. We would appreciate specific guidance on which aspects of diversity measurement need additional clarification.
> > >
> > > [1]: https://www.researchgate.net/figure/Distribution-of-document-lengths-in-Pile-The-highest-1-percentile-of-document-length-are_fig5_348212145

---

### Comment · Reviewer_VhJ6 · 2024-12-02

I believe I have not heard back individually from the authors, and the review period is coming to a close. It would be good to hear back and have a response from the authors' given there is not much time remaining for review and discourse.

---

### Meta-Review · Area_Chair_kkAK · 2024-12-10

**Metareview:**

Reject by reviewer consensus.

The paper introduces EXE, a benchmark designed to evaluate language models (LLMs) on their ability to execute Python code sourced from github and with LLM generated test cases. Reviewers generally understood this work and liked the motivation as well as how it represents real-world Python code.

Several reviewers think that only evaluating on GPT-* models is insufficient and urged inclusion of at least some open models. Reviewers also pointed out various issues such as diversity (only Python), and whether model generated test cases are trustworthy.

**Additional Comments On Reviewer Discussion:**

Reviewers are in agreement. Authors responded very late to reviewer comments, not much discussions needed.

---

### Decision · Program_Chairs · 2025-01-22

Reject